# Identifying control ensembles for information processing within the cortico-basal ganglia-thalamic circuit

**Catalina Vich**[1,2], **Matthew Clapp**[3,4], **Jonathan E. Rubin**[4,5☯]*, **Timothy Verstynen**[3,4☯]*

**1** Dept. de Matemàtiques i Informàtica, Universitat de les Illes Balears, Palma, Spain, **2** Institute of Applied Computing and Community Code, Palma, Spain, **3** Department of Psychology & Neuroscience Institute, Carnegie Mellon University, Pittsburgh, Pennsylvania, United States of America, **4** Center for the Neural Basis of Cognition, Pittsburgh, Pennsylvania, United States of America, **5** Department of Mathematics, University of Pittsburgh, Pittsburgh, Pennsylvania, United States of America

☯ These authors contributed equally to this work.
* jonrubin@pitt.edu (JR); timothyv@andrew.cmu.edu (TV)

**Data Availability Statement:** All simulation and analysis code reported in this work is publicly available at: https://github.com/CoAxLab/CBGTControlEnsembles.git.

## Abstract

In situations featuring uncertainty about action-reward contingencies, mammals can flexibly adopt strategies for decision-making that are tuned in response to environmental changes. Although the cortico-basal ganglia thalamic (CBGT) network has been identified as contributing to the decision-making process, it features a complex synaptic architecture, comprised of multiple feed-forward, reciprocal, and feedback pathways, that complicate efforts to elucidate the roles of specific CBGT populations in the process by which evidence is accumulated and influences behavior. In this paper we apply a strategic sampling approach, based on Latin hypercube sampling, to explore how variations in CBGT network properties, including subpopulation firing rates and synaptic weights, map to variability of parameters in a normative drift diffusion model (DDM), representing algorithmic aspects of information processing during decision-making. Through the application of canonical correlation analysis, we find that this relationship can be characterized in terms of three low-dimensional control ensembles within the CBGT network that impact specific qualities of the emergent decision policy: responsiveness (a measure of how quickly evidence evaluation gets underway, associated with overall activity in corticothalamic and direct pathways), pliancy (a measure of the standard of evidence needed to commit to a decision, associated largely with overall activity in components of the indirect pathway of the basal ganglia), and choice (a measure of commitment toward one available option, associated with differences in direct and indirect pathways across action channels). These analyses provide mechanistic predictions about the roles of specific CBGT network elements in tuning the way that information is accumulated and translated into decision-related behavior.

**Funding:** CV received support through a PCI2020-112026 project, funded by MCIN/AEI/10.13039/501100011033 (https://www.ciencia.gob.es/Ministerio/mision-y-organizacion/Entidades-Adscritas/agencia-estatal-de-Investigacion-aei.html) and the European Union "NextGenerationEU"/PRTR ("NextGenerationEU"/PRTR). TV and JER received funding from the National Institutes of Health award R01DA053014 (https://www.nih.gov). These are all part of a joint project under the CRCNS program (http://crcns.org). The funders had no role in study design, data collection and analysis, decision to publish, or preparation of the manuscript.

**Competing interests:** The authors have declared that no competing interests exist.

## Author summary

Mammals are continuously subjected to uncertain situations in which they have to choose among behavioral options. The cortico-basal ganglia-thalamic (CBGT) circuit is a complicated collection of interconnected nuclei believed to strongly influence the ability to adapt to environmental changes. The roles of specific CBGT components in controlling information during decisions remain unclear. At a more phenomenological, algorithmic level, drift-diffusion models have been shown to be able to reproduce behavioral data (action selection probabilities and the time needed to make a decision) obtained experimentally from mammals and to provide an abstract representation of a decision policy. In this work, we use simulated decision-making to establish a mapping from neural activity in the CBGT circuit to behavioral outcomes. This mapping illuminates the importance of three core sets of CBGT subnetworks in the action selection process and how they are involved in adapting decision policies across exploitative and exploratory situations.

## Introduction

Although making a decision can feel instantaneous, decisions in fact arise gradually from the ongoing processing of external (e.g., sensory) and internal (e.g., learned contingencies) information streams. In this process, the inclination toward selecting one action over others is continually updated until sufficient evidence is reached to allow one action, or a set of actions, to proceed [1]. The parameters associated with this process, such as the speed of integration of incoming information and the level of evidence needed to make a decision, define the decision policy [2]. Shortening the window of time available for evidence accumulation often, although not always, leads to faster but more random or exploratory decisions. Lengthening the time for evidence accumulation often leads to slower and more "greedy", or exploitative, decisions. A fundamental challenge for the brain is to manage this speed-accuracy tradeoff, and the associated exploration-exploitation dilemma, via management of the evidence accumulation process, using both current context and prior experience to promote effective outcomes for any given situation [3].

A likely control center for information processing during decisions is the cortico-basal ganglia-thalamic (CBGT) network (see Fig 1 upper panel). The canonical CBGT circuit includes two structurally and functionally dissociable control streams: the *direct* (facilitation) and *indirect* (suppression) pathways [4]. Central to the canonical model is the assumption that the basal ganglia are organized into multiple action channels [5–9], each containing direct and indirect pathway components. While in reality actions are likely represented in a less thoroughly segregated fashion across CBGT circuits [10, 11], the concept of independent action channels provides conceptual ease when describing the competition between possible actions without changing the key dynamic properties of the underlying computations. The classical view of these pathways [4, 12, 13] is that activation of the direct pathway, via cortical excitation of D1-expressing spiny projection neurons (SPNs) in the striatum, unleashes GABAergic signals that can suppress activity in the CBGT motor output nuclei (e.g., predominantly the internal segment of the globus pallidus, GPi, in primates and the substantia nigra pars reticulata, SNr, in rodents) and hence relieve the thalamus from the tonic inhibition that basal ganglia outputs normally provide. This release from inhibition allows the thalamus to facilitate action execution. Conversely, activation of the indirect pathway, via D2-expressing SPNs in the striatum, can alter firing in the external segment of the globus pallidus (GPe) and the subthalamic nucleus (STN) in a way that strengthens basal ganglia inhibitory output. This result suppresses

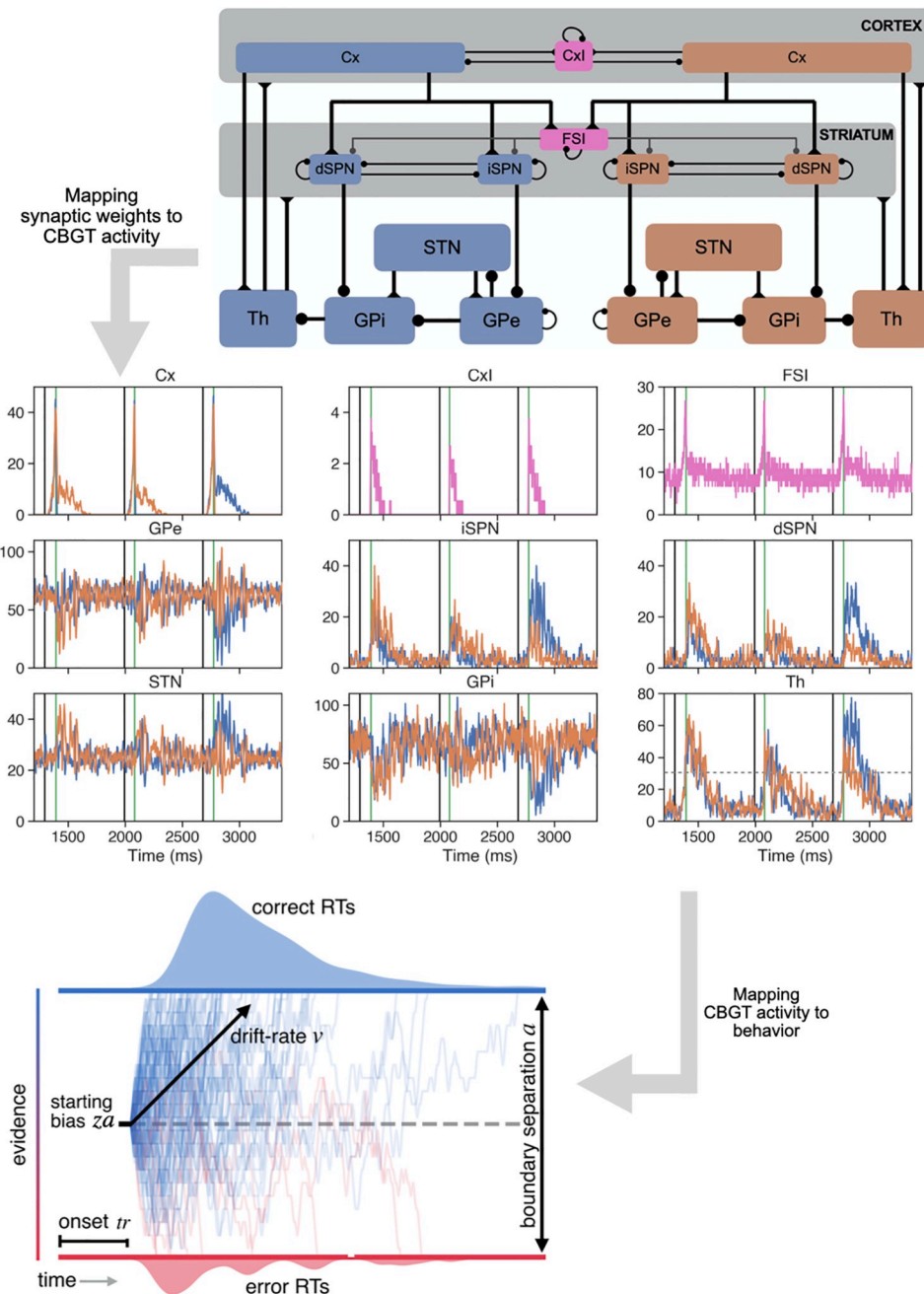

**Fig 1. Mapping cortico-basal ganglia-thalamic (CBGT) circuitry to behavioral data.** The CBGT network includes neurons from a variety of interconnected populations (see Section Neural activity: CBGT network). Blue nodes and orange nodes represent two distinct action channels, which we call A and B; pink nodes are those populations that interact with both channels. Connection weights, *W*, in the CBGT network (upper panel) modulate network firing rates, *R*, (central panel) that in turn map to behavior that can be fit by a DDM (lower panel) after tuning of its parameters, *P*. In the central panel, black vertical traces indicate the stimulus starting times for three decision-making trials, while the green vertical lines mark the times when the decisions (B,B,A) were made. The horizontal dashed gray line in the thalamus (Th) subpanel corresponds to the decision threshold of 30 *Hz*. Cx, cortical cells; CxI, cortical inhibitory interneurons; FSI, fast spiking interneurons; dSPN, direct pathway spiny projection neurons; iSPN, indirect pathway spiny projection neurons; STN, subthalamic nucleus; GPe, external globus pallidus; GPi, internal globus pallidus; Th, thalamus.

activity of motor pathways and reduces the likelihood of action selection. More recent experiments have revealed nuances of pathway interactions that go beyond these principles. For example, recordings show that both indirect and direct pathway SPNs can increase their activity together in the lead-up to a decision [14–16]. The topological encoding of actions in the striatum [10], along with the convergence of projections to the GPi/SNr [17, 18], also suggests that the direct and indirect pathways may compete for control over the output of the basal ganglia, encoding the "evidence" favoring any behavioral decision as the relative activation of the two pathways *within* the corresponding action channel [19–21]. Overall, the subtleties of this interaction across pathways and the details of how decision policies emerge from CBGT circuits remain to be determined.

Over the past decade an alternative view has emerged, which posits that the primary functional role of CBGT circuits is to control movement vigor [22, 23]. According to this model, rather than serving as a gate to decisions, the opponent activity of direct and indirect CBGT pathways controls the vigor and speed with which movements are executed. For example, optogenetic stimulation of direct or indirect striatal SPNs in mice increases or decreases, respectively, the speed of goal directed movements [24]. During motor learning, striatal SPNs tune their responses to the required kinematics of the task, most prominently running speed [25]. In non-human primates performing a reach task featuring a speed-accuracy trade-off, pallidal activity was found to signify urgency and to impact the vigor with which selected movements were performed [26]. Finally, human patients with pallidal lesions are able to adequately scale their voluntary grip force when holding an object, but lack the spontaneous force calibrations necessary for maintaining a grip on objects, a phenomenon taken to reflect a lack of vigor in stabilizing movements [27].

We argue that the CBGT network's role in action selection and its role in control of movement vigor can be reconciled by understanding precisely how the pathways within the CBGT control information as it progresses through the network. Specifically both the action selection and vigor perspectives focus on the speed, or energy, at which signals propagate through the circuit, suggesting that they both rely on control of information propagation through CBGT networks. Within the high-dimensional space of CBGT circuits, there exists a likely lower-dimensional mapping between the activity of specific subnetworks and specific properties of the evidence accumulation process. Here we refer to these subnetworks as *control ensembles* and the corresponding decision properties that they influence as *factors*. Thus, synaptic plasticity, neuromodulation, and other top-down control signals that result in changes in the activity of control ensembles will adjust factors and hence affect the behavioral outcomes that arise when using sensory signals to guide motor decisions. In past computational work, we showed how plasticity-induced changes in direct-vs-indirect pathway influence could lead to changes in two specific decision factors linked with parameters of the accumulation of evidence process during decision-making [28]: the rate of information accumulation (related to the difference in direct pathway activity between channels) and the threshold of evidence needed to initiate an action (tuned by overall indirect pathway activity, see also [29, 30]). This prior study suggests that the specific synaptic configuration of CBGT pathways, which shapes the firing dynamics of neurons in CBGT subpopulations, may tune certain factors of the decision policy by controlling the way that information is used to guide future decision-making.

In this work, we move beyond this preliminary idea to develop a much more complete mapping and understanding of the possible control ensembles within CBGT circuits that drive specific information processing factors. To do so, we first sought to determine what possible tunings of synaptic weights within CBGT pathways would result in firing rates compatible with experimental observations, and hence are biologically reasonable to consider. Second, we identified how the modulation of the individual weights within these ranges would impact

**Table 1. Number of neurons in each population.** The numbers corresponding to CxI and FSI are the total number of neurons. The other values correspond to neurons per channel.

| Population | CxI | Cx | dSPN | iSPN | FSI | GPi | GPe | STN | Th |
|---|---|---|---|---|---|---|---|---|---|
| Number of neurons | 186 | 204 | 75 | 75 | 75 | 75 | 750 | 750 | 75 |

decision-related parameters, and hence could serve as building blocks of the control ensembles and overall network tunings that implement various decision policies. For the latter step, we ran simulations in which basal ganglia output firing rates in two competing action channels were used to determine decision outcomes and response times. We fit the distributions of these simulated behavioral variables with the drift diffusion model (DDM) [2, 31], a canonical formalism for the process of evidence accumulation during decision-making. We subsequently used canonical correlation analysis (CCA) to compute a low-dimensional mapping between CBGT synaptic weights and DDM decision parameters. CCA yielded three collections of weights, each of which provides the strongest impact on a corresponding component of the DDM parameter vector and hence acts as the control ensemble for that factor of the decision-making process. This analysis highlights how the CBGT pathways encode multiple mechanisms that control the behavior associated with decision-making processes, some that regulate vigor and some that regulate choice, which collectively work together to manage the speed-accuracy tradeoff during movement selection and control.

## Results

### CBGT network dynamics and behavior

Our main goal in this work was to establish a mapping between CBGT properties and decision parameters, thereby identifying potential control ensembles within CBGT networks. To this end, we simulated a spiking model for the CBGT network that included two action channels (Fig 1, upper panel; see also Table 1 and other details in the Methods section titled Neural activity: CBGT network) and declared an action to be selected when the instantaneous firing rate of the thalamic population for a channel first reached a pre-specified decision threshold (taken for concreteness as 30 Hz [28]). Before we can progress to discussing control ensembles and decision factors, however, we first need to demonstrate that our simulations a) produced realistic dynamics that qualitatively map to known biological observations and b) exhibited a dynamic range of behavior when we performed directed sampling of the underlying synaptic weights. With regard to the dynamics, the CBGT network produced average firing rates across neurons in each population during baseline periods as well as during decision-making that agree with those observed experimentally [32–40], as documented in Table 2, when the network was integrated with the control conductances $g$ given in Table 3 (2nd column) and with an external input presented simultaneously to the cortical populations $Cx$ for both the A and B channels (see Methods section Neural activity: CBGT network). The mean firing rates of all populations as a function of time over a sequence of three actions (B, B, A) are depicted in Fig 1 (central panel).

A clearer picture of how firing rates evolve over the course of deliberation is obtained by averaging over many trials (Fig 2). At the start of each trial, a constant stimulus to both cortical sensory models was turned on; because we had no difference in inputs to the populations $Cx_A$, $Cx_B$, both actions had an equal chance of being selected and the firing rates in the A and B subpopulations for each CBGT region were similar, albeit with small differences due to randomness in the selection of initial conditions. This sustained input resulted in a nonlinear increase in cortical firing rates (Fig 2, upper left) as well as in firing rates across striatal populations

**Table 2. Firing rate ranges during baseline and during decision tasks for different populations in the brain.** These ranges reflect experimental data from both primates and rats.

| Population | baseline FR range (Hz) | full FR range (Hz) | References |
|---|---|---|---|
| dSPN | [0, 5] | [0, 35] | [32–35] |
| iSPN | [0, 5] | [0, 35] | [32–35] |
| GPe | [40, 90] | [40, 150] | [36–38] |
| GPi | [40, 90] | [40, 150] | [38] |
| STN | [10, 35] | [10, 55] | [36–38] |
| Th | [5, 20] | [5, 85] | [39] |
| Cx | | [0, 100] | [40] |
| FSI | [5, 40] | [5, 70] | [40] |

**Table 3. CBGT connectivity.** From left to right, the connections included, specified in terms of the pre- and the post-synaptic populations that they link; the probability that two cells in the pre- and post-synaptic populations will be synaptically linked; a fixed synaptic conductance or weight (measured in *nS*) used in our control case; the lower and upper bounds on weights based on maintaining realistic neuronal firing rates; and finally the receptor types for each specific connection. Those connections that are not in bold font (i.e. without an upper and lower bound) are fixed to the control value during all simulations. The topology of each connection, which can be either diffuse (they project to left and right action channels) or focal (restricted connections within each channel), has not changed from [28](Table 3).

| Connection type | Connection Probability | g(*nS*) control | Lower g bound | Upper g bound | Receptor(s) |
|---|---|---|---|---|---|
| Cx-Cx | 0.43 | 0.0127 | | | AMPA |
| Cx-Cx | 0.43 | 0.15 | | | NMDA |
| Cx-CxI | 0.2417 | 0.113 | | | AMPA |
| Cx-CxI | 0.2417 | 0.525 | | | NMDA |
| CxI-Cx | 1.00 | 1.75 | | | GABA |
| CxI-CxI | 1.00 | 3.5833 | | | GABA |
| **Cx-dSPN** | 1.00 | 0.027 | 0.012 | 0.033 | NMDA |
| **Cx-iSPN** | 1.00 | 0.027 | 0.0195 | 0.0885 | NMDA |
| **Cx-dSPN** | 1.00 | 0.018 | 0.008 | 0.022 | AMPA |
| **Cx-iSPN** | 1.00 | 0.0986 | 0.008 | 0.059 | AMPA |
| **Cx-Th** | 1.00 | 0.035 | 0.008 | 0.0548 | NMDA, AMPA |
| **Cx-FSI** | 1.00 | 0.198 | 0.1905 | 0.63 | AMPA |
| dSPN-dSPN | 0.45 | 0.28 | | | GABA |
| dSPN-iSPN | 0.45 | 0.28 | | | GABA |
| **dSPN-GPi** | 1.00 | 2.09 | 0.418 | 2.413 | GABA |
| iSPN-dSPN | 0.5 | 0.28 | | | GABA |
| iSPN-iSPN | 0.45 | 0.28 | | | GABA |
| **iSPN-GPe** | 1.00 | 4.07 | 2.47 | 4.46 | GABA |
| **GPe-STN** | 0.067 | 0.35 | 0.33 | 0.39 | GABA |
| GPe-GPe | 0.067 | 1.75 | | | GABA |
| **GPe-GPi** | 1.00 | 0.06 | 0.05733 | 0.067 | GABA |
| **GPi-Th** | 1.00 | 0.3315 | 0.32017 | 0.357 | GABA |
| FSI-dSPN | 1.00 | 1.7776 | | | GABA |
| FSI-iSPN | 1.00 | 1.66987 | | | GABA |
| FSI-FSI | 1.00 | 3.2583 | | | GABA |
| **STN-GPe** | 0.1617 | 0.07 | 0.05 | 0.1 | AMPA |
| STN-GPe | 0.1617 | 1.51 | | | NMDA |
| **STN-GPi** | 1.00 | 0.038 | 0.036 | 0.03833 | NMDA |
| **Th-Cx** | 0.83 | 0.03 | 0.021 | 0.035 | NMDA |
| Th-CxI | 0.83 | 0.015 | | | NMDA |
| **Th-dSPN** | 1.00 | 0.3825 | 0.0015 | 0.3915 | AMPA |
| **Th-iSPN** | 1.00 | 0.3825 | 0.3525 | 0.3975 | AMPA |
| Th-FSI | 0.83 | 0.1 | | | AMPA |

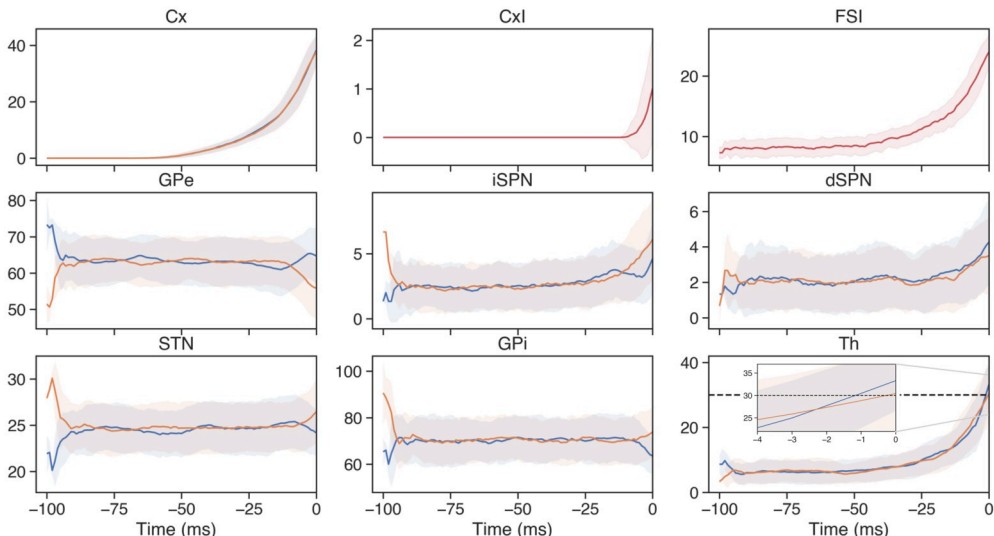

**Fig 2. Mean firing rates of the populations in the CBGT network on choice A trials.** The firing rate in spikes/s is averaged across all trials for which action A was selected with the control set of synaptic conductances (third column in Table 3), from the stimulus onset time at *Cx* to the decision time (set to be 0). Blue traces correspond to the sub-populations affecting the A action, orange traces represent the sub-populations impacting the B action, and red traces are used for the populations (*CxI* and *FSI*) common to both action channels. Shadow areas represent the standard deviation for each population. Since each trial had a different reaction time, we performed the averaging by aligning trials on their decision times. Same cell type abbreviations as in Fig 1.

(dSPN, iSPN, FSI). The inhibition of SPNs by the higher-firing FSIs allowed FSI activity to rise earlier than SPN firing, but eventually the SPNs did follow the cortical surge. The firing rates of the other populations also remained steady throughout much of the period leading up to an action. In particular, the flat GPi firing rates indicate that its inhibitory inputs from dSPNs and GPe and its excitatory inputs from STN remained balanced, but eventually with this set of conductances and cortical drive, the balance broke, GPi firing decreased in the channel for the selected action, and thalamic firing in that channel quickly rose to the decision threshold (cf. [20, 28]).

In our simulations of the baseline network parameters, the synaptic conductances were identical in the A and B channels. Consequently, we expect that the network would have chosen actions A and B with similar likelihood on each trial and with similar frequencies across many trials. Consistent with this assumption, using the baseline parameters, over the course of 300 trials action *A* was selected 45.3% of the time, with an average reaction time of 86.45 *ms* across all trials (86.56 *ms* for A actions, 86.36 *ms* for B actions).

By independently varying the synaptic conductances of many of the connections within the network, but maintaining the same value for each conductance across the two channels, we determined a range of conductance values (see Table 3, 4th and 5th columns) over which neuronal firing rates remained within our pre-specified acceptable ranges, based on published experimental recordings from these populations (Table 2). This gave us a sampling window for each synapse that would still produce relatively stable and realistic firing rates. We note that these do not represent the largest ranges of values that would preserve these characteristics. Indeed, after we established these maximal ranges, we used them to perform Latin hypercube sampling (LHS; see Section Network specification and network behavior) and thus to generate 300 different configurations of network weights. For each sampled network, we simulated 300 choices (i.e., trials). Initially, we found that simultaneously varying parameters within

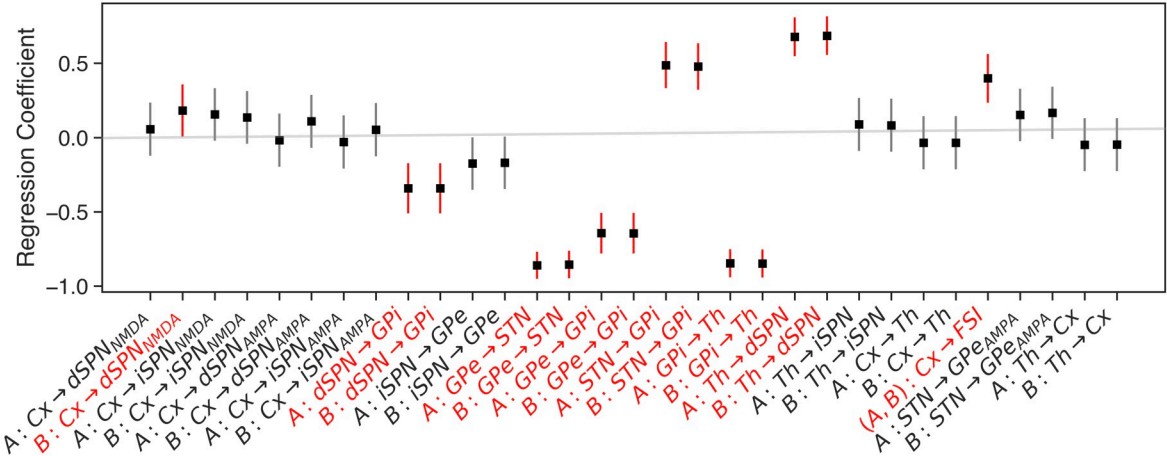

**Fig 3. Influence of synaptic weight variation on corresponding postsynaptic firing rates.** Black dots represent the estimated regression coefficient for a linear regression model with sampled synaptic weight as an input and post-synaptic unit firing rate as the output. Vertical lines correspond to the 95% confidence interval computed using the standard error of the estimate (gradient) under the assumption of residual normality. Grey error bars correspond to those cases that include the 0 regression coefficient for the slope while red error bars are the others. Both weights and firing rates were z-scored before the analysis.

the LHS procedure could push some firing rates out of bounds. Thus, we completed an iterative process of successively shrinking our acceptable ranges of synaptic conductance values and performing LHS until the average pre-selection firing rates were preserved within our pre-specified ranges (see S1 Fig).

We next examined the impact of each synapse on the firing rate of the corresponding post-synaptic population, based on the full set of results obtained from varying synaptic conductances via the LHS procedure. Note that the relationship between synaptic weight and postsynaptic rate is not as simple as it sounds, because of the multiplicity of pathways impacting the firing of many populations within the CBGT (Fig 1, upper panel). Fig 3 shows the degree to which changes in synaptic weights influenced the firing of the post-synaptic neurons, averaged across trials over the full period from trial onset to decision time. This association is represented as a simple linear regression coefficient, representing the slope of the least squares linear relation between the weight and the rate. The first observation that arises from this calculation is that many post-synaptic population firing rates did not exhibit systematic variation under changes in synaptic weights. Indeed, while our LHS procedure effectively changed local firing rates of post-synaptic cells, the magnitude and linearity of this influence varied substantially across pathways. Second, the directions of influence matched the expected changes given the nature of the synaptic connections involved: increases in inhibitory synapse strength largely yielded decreases in postsynaptic firing rates, while increases in excitatory synapse strength induced increases.

Each configuration of weights used in the LHS procedure created a unique network with specific dynamics (i.e., firing rates) and information processing properties. Example results from the 300 trials performed with a single network are shown in Fig 4. This configuration produced a network with a fairly unbiased selection probability, and an average RT near the modal value observed across all network configurations (see Fig 5 below). Pre-decision firing rates for this network, across all trials, also stayed well within biologically realistic ranges.

Overall, the LHS procedure yielded an approximately unimodal, symmetric distribution of average reaction times spread over a relatively broad range, with greatest concentration between 50 and 150 *ms* (Fig 5A). This variation is largely consistent with the range of

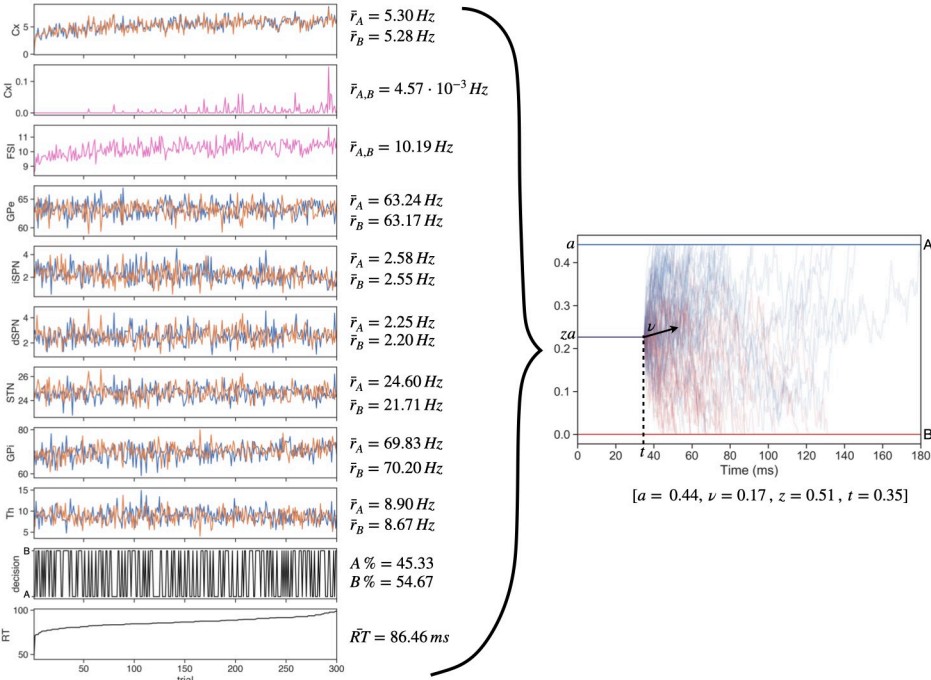

**Fig 4. Example of data generated for a single network configuration.** The left panels show the the average pre-decision firing rates, decision outcomes (1 = A, 0 = B), and reaction times across all 300 trials of a single network instance. Mean values are reported next to each plot. From the network behavior a single set of DDM parameters was fit; these parameter values are listed in the right panel, which shows a collection of DDM runs with these parameters. These values represent one sample in the distributions shown in Fig 5. Trials are sorted in order of increasing reaction time.

variability seen in humans (see [41]) and rodents (see [42]) during time-constrained RT tasks. That is, while the mean reaction times in our simulations were shorter than what is typically observed in both humans and non-human animals, the width (i.e., variance) of the distribution is similar to empirical observations. It is important to note that we were not simulating either sensory processing time or motor planning and execution processes, which would add approximately 100–200 *ms* to the response time in a full model system (for example, see [43]). Indeed, the reaction times of our network, which reflect when the *cortex* makes a decision, are in line with onset times of cortical ramping activity before movement onsets (for example, see [44]). As far as the actions selected, across LHS samples, the percentage of A choices remained close to 50% (Fig 5B), reflecting a relatively unbiased sampling procedure.

The modulation of both choice and reaction time across parameter variations (i.e., unique tunings) of the network, despite *identical* sensory signals across cases, suggests that the LHS sampling effectively impacts how information is used in the deliberation process. To understand how parameter variations modulate the components of this process, we fit each network's choice and reaction time distributions to a hierarchical drift-diffusion model (HDDM, [31]). This procedure returned four separate information processing parameters—boundary height $a$, drift rate $v$, onset time $t$, and starting bias factor $z$ (Fig 1, bottom; see also Methods section, and note that for implementation purposes, the actual bias implemented in the DDM is $z \times a$, as shown in the DDM plot in Fig 4)—for each version of the network. These summary measures, namely average firing rates, selection behavior, and estimated DDM parameters, were saved for each network for further analysis. Importantly, in each case, goodness-of-fit

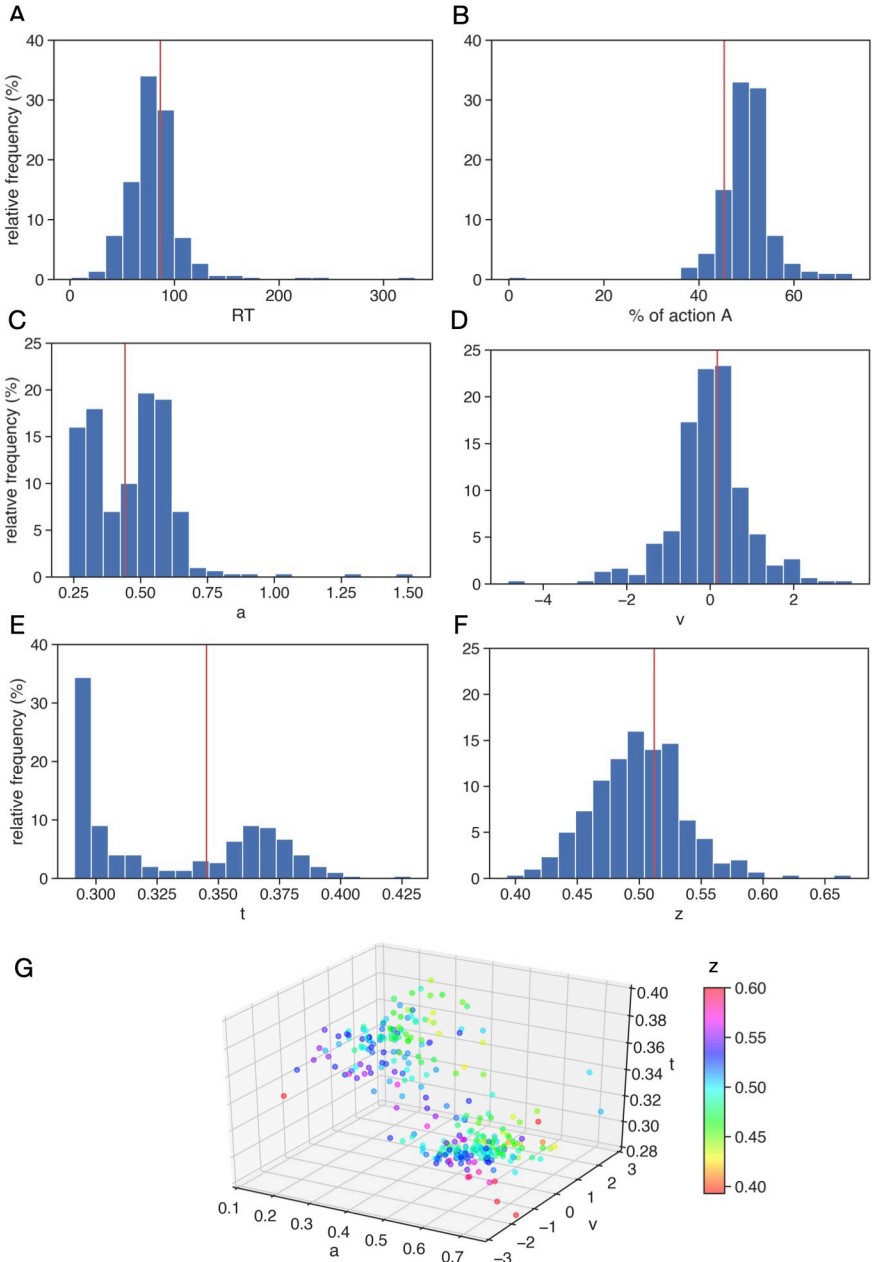

**Fig 5. Histograms of simulated behavioral data and best fit DDM parameter values.** Panels A and B display the histograms of the reaction times (RT, averaged over 300 trials) and the percentage of choice A (also averaged over 300 trials) across 300 networks generated via the LHS procedure. For each network, the distribution of RTs and accuracy on each trial were used to estimate the DDM parameters ($a$, $v$, $t$, $z$), which are represented in histograms in panels C-F. The vertical red lines in panels A-F correspond to the specific value obtained for the permuted network based on the control conductances (Fig 4). Panel G is a 3D representation of the ($a$, $v$, $t$) parameter space with parameter $z$ represented by the colorbar, which is capped at 0.6 (despite one outlier at $z > 0.65$ and another at $z \in (0.6, 0.65)$; see panel F). In this panel, each dot represents the ($a$, $v$, $t$, $z$) vector obtained using the HDDM algorithm for a specific sampled network.

was evaluated by computation of the deviance information criterion (DIC); see S4 Fig. DIC values in all but four of the 300 configurations indicated a good fit (comparable to best fit cases in our past work [45]); subsequent analyses were performed both with and without the four poor-fit cases and produced essentially identical results.

Histograms indicating the distributions of the estimated DDM parameters ($a$, $v$, $t$ and $z$) across all sampled networks are shown in Fig 5C–5F. The first pattern that pops out in these distributions is the presence of bimodality across the $a$ and $t$ parameters, while in contrast the distributions for the drift rate $v$ and the bias factor $z$ are unimodal. In particular, the distribution for $v$ is appropriately centered at 0, which corresponds to an absence of drift towards either option as is suitable for the simulated scenario that lacks the differences in reward or input that would normally impart a direction to the drift rate. The bimodal distributions observed for $a$ and $t$ are related to each other and reflect a compensation, as can be seen from Fig 5G. Specifically, the concentration of relatively high $a$ values, which would promote slower decisions, corresponds to relatively low $t$ values, which would accelerate decisions, and vice versa. Moreover, we find a second compensatory relationship in that the largest bias factors, $z$, arise in combination with strong negative drift rates, within the particle cloud with relatively large $t$. In contrast, lower $z$ values are positively associated with large values of $v$. This suggests the possibility that the specific tunings generated by our LHS procedure give rise to two distinct types of general approaches to decision-making: early onsets of evolution of the process with elevated boundary heights and later onsets with lower boundary heights. Since the directions of the changes between these two parameters have countermanding influences on response speed, the net speed remains approximately conserved across these two clusters of decision policies. We also note that although values of $a$ and especially $t$ seem to accumulate at lower boundaries of their distributions, this effect emerged naturally, as no corresponding bound was present in the fitting process.

To quantify the clusters that we see visually in Fig 5G, we applied $K$-means clustering to the DDM parameters, across network permutations, and computed the silhouette coefficient $s$ (see Methods section $K$-means clustering). We found that the best silhouette coefficient is reached for $K = 3$ with value $s = 0.68$, meaning that the DDM parameter space can be clustered into three different groups of parameter values: two subsets that separate high and low values of the onset time and the boundary height, and a third subset that only contains a small set of data points with very high boundary heights that we treat as outliers. As shown in S2 Fig, the aforementioned separation splits the bimodal distributions observed for $t$ and $a$ in Fig 5E and 5F, with each cluster effectively representing a different strategy for producing a similar set of RTs and accuracy in the DDM. This analysis largely confirms the clustering patterns we identified visually in the parameter distributions.

We also repeated the fitting process 50 times for each of eight randomly selected CBGT network configurations, four for which the original DDM parameters belong to one of the main clusters and four for which they belong to the other main cluster. All of the fits obtained belonged to the same cluster as the original fit, confirming that the clusters are a true feature of the best fits to CBGT outputs and not a figment of bistability in the fitting landscape. We note that although the fits for parameter values in both of the main clusters gave small DIC values (S4 Fig), the data corresponding to the cluster with larger values of $t$ and smaller values of $a$ generally yields lower (more negative) DIC values over the common range of reaction times captured by both clusters. Nonetheless, the two clusters together can yield a broader range of RTs than arise in either cluster individually, and each cluster represents the decision policy for a given CBGT network configuration. Thus, despite the difference in DIC distributions between clusters, we include the fits for both clusters in our subsequent analyses.

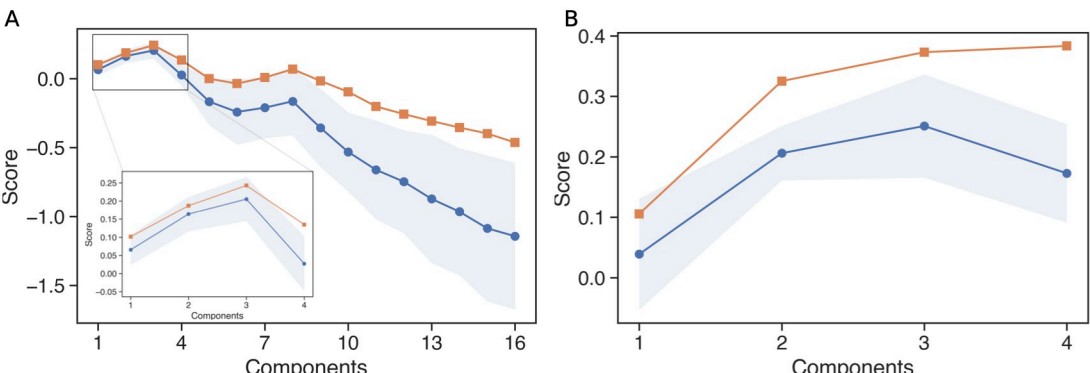

**Fig 6. Scores obtained when different numbers of components are considered in the CCA.** Panel A represents the scores for the CCA considering the data sets $W$ and $R$ while panel B shows the scores when using $R$ and $P$. Note that only 4 components are considered for the latter because the DDM model only features 4 parameters. Blue traces depict the mean scores over the 4-fold cross-validation analysis (see Methods, Canonical Correlation Analysis) while the orange traces show the scores for the full data sets (no cross-validation analysis performed). The shadowed blue zone corresponds to the standard deviation of the scores in the cross-validation analysis.

## Low-dimensional control ensembles

We used canonical correlation analysis (CCA, see Methods section Data analysis and [46]) to uncover the low-dimensional relationships between network features and decision parameters that arise as we move up effective levels of analysis in the model. We ran two CCA evaluations, one exploratory and one to address our central hypothesis that specific forms of modulation of CBGT firing rates can tune behavioral factors; these were a first instance to map synaptic weight schemes $W$ to CBGT firing rates $R$ ($\mathcal{G} : W \rightarrow R$; exploratory evaluation) and a second instance to map $R$ to the DDM parameters $P$ ($\mathcal{H} : R \rightarrow P$; hypothesis evaluation). For each evaluation, a 4-fold cross-validation analysis (see Methods section Data analysis) on each of the data sets $W \rightarrow R$ and $R \rightarrow P$ showed that a 3-component model best explained variance for $W \rightarrow R$ and another 3-component model best explained variance for $R \rightarrow P$ (Fig 6, peaks at 3 in blue curves). Specifically, the best model for the $W \rightarrow R$ test explained 20.8% of the variance in the hold out test data, with a standard deviation of 5%. The best model for the $R \rightarrow P$ test explained 26.9% of hold out test variance, with a standard deviation of 7%.

**Weights to firing rates.** The exploratory CCA for $W \rightarrow R$ did not reveal any surprising effects and was dominated by a few specific pathways (see S3 Fig for details). Based on the leading component, the strongest relationship observed represents the straightforward, expected finding that strengthening the inhibitory synapse from GPi to thalamus associates with decreasing thalamic firing rates. The second component resolves a point of minor ambiguity: since GPe and STN form a reciprocal loop, the effects of changing weights of one connection pathway between the two areas are hard to predict, but here we find that increases in the inhibition from GPe to STN associate with decreases in activity in STN and, presumably through the resulting loss of excitatory input, in GPe as well. Interestingly, the third component represents a higher order form of the same relationship found in the first component: it shows that synaptic changes favoring inhibitory over excitatory inputs to the GPi associate with decreasing GPi firing rates and increasing thalamic firing rates. Put together, this exploratory CCA model illustrates that among all of the weight changes implemented in our parameter exploration, variation in the weights for the primary output nodes of the basal ganglia contribute the most to variability in firing rates. Given, however, the recurrent architecture of the network

and its nonlinear properties, more subtle contributions of other synaptic connections to firing rates may simply be washed out by the dominance of GPe, STN, and GPi influences on thalamic nuclei.

**Firing rates to decision parameters.** The CCA model evaluating our primary hypothesis, that of a relationship in $R \rightarrow P$, proved to be more illuminating. For this analysis firing rates were grouped in two ways, based on measures of striatal activity found to be most influential in our previous work [45]: 1) overall activity of populations in a CBGT region averaged across channels A and B, and 2) the difference in firing rates in the region between its channel A and channel B populations. Note that a nice feature of these rate groupings is that, unlike single channel measures, they reveal patterns that occur across trials regardless of which decision was made. Fig 7 shows color-coded representations of the entries, or loadings, of the matrix **U** composed of the firing rate components and of the matrix **V** comprising the DDM parameter components from the CCA for $R \rightarrow P$ (see Methods section Data analysis), with overall firing rates shown in the top half and between-channel differences in firing rates shown in the bottom half of rows in Fig 7A. The loadings of each firing rate component can be interpreted as coefficients in a linear combination of the indicated firing rate measures, representing a vector **u** in the corresponding 16-dimensional space, and the loadings of each corresponding DDM parameter component as coefficients in the linear combination of these quantities that represents the vector **v** in DDM parameter space most strongly correlated with **u**. The three (**u**, **v**) pairs identified represent the most strongly correlated vector pairs identified from these two spaces.

Comparing across columns, we see a clear pattern emerge with three unique components, or factors, in the decision policy (i.e., DDM) space. The first factor has a strong, negative loading on onset time, $t$, and a weaker negative loading on boundary height, $a$. This pattern is consistent with a *responsiveness* influence on the decision policy, modulating how quickly evidence evaluation begins and the overall response speed, with no definitive bias towards one action or another. The second factor, by contrast, manifests as a strong positive loading on $a$ and a slightly weaker negative loading on $t$. This pattern is consistent with a *pliancy* component, modulating the degree of evidence necessary to make a decision (i.e., length of evidence accumulation vector) by primarily impacting the domain size for the diffusion process, but not its direction. The final factor is expressed as a positive loading on drift rate, $v$, with a weaker negative loading on the bias factor, $z$. This pattern is consistent with an action *choice* component, modulating the likelihood of selecting a specific option by impacting the direction of the

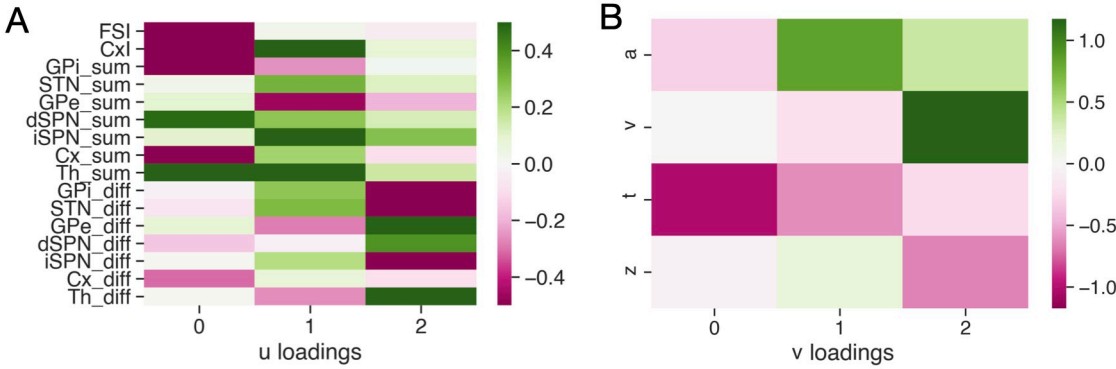

**Fig 7. Canonical loading matrices obtained by applying the CCA to the mapping $R \rightarrow P$.** A) The loadings corresponding to the firing rates, $R$. B) The loadings corresponding to the DDM parameters, $P$. In both panels, the values of matrix entries are represented by the color bars, and the color code indicates the magnitude and sign of the loading.

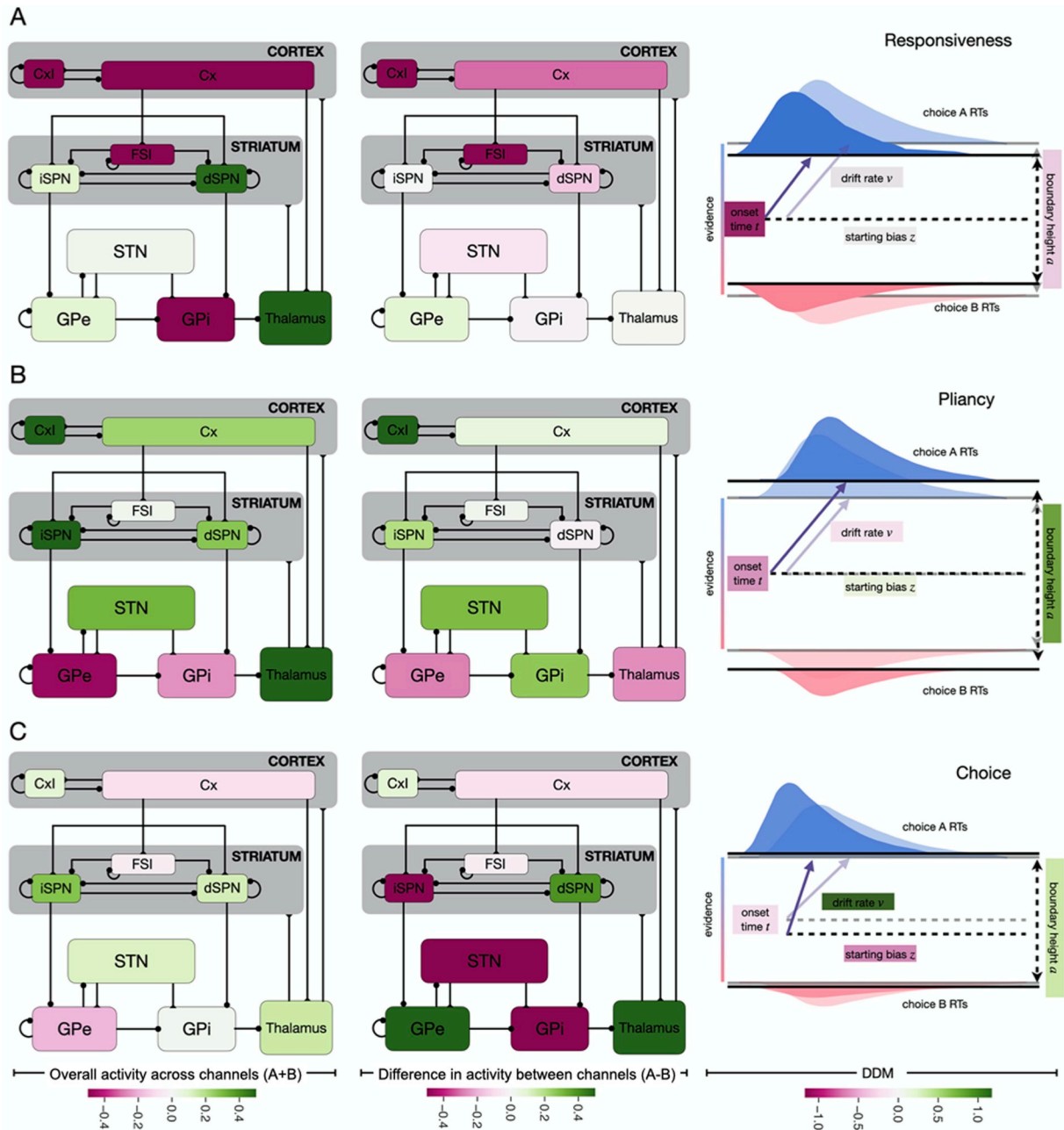

**Fig 8. Canonical loadings obtained by applying the CCA to the mapping $R \rightarrow P$.** Panels A, B, and C represent the relation between the firing rate of each population in the CBGT network and the parameters of the DDM obtained in the first, second and third components of the CCA, respectively. 1st column subpanels represent the overall activity summed across both channels while the 2nd column represents the differences in activity between the populations in the A and B channels. 3rd column subpanels depict the associated increases (green) and decreases (magenta) in the DDM parameters.

diffusion process. We will examine each of these components in turn. We visualize the factors as they are manifested in the CBGT network and the DDM, using the same color coding from Fig 7, in Fig 8.

The *responsiveness* factor was largely associated with the overall activity of corticothalamic systems and the direct pathway projections (Fig 8A). As expected from the logic of the circuit,

greater activity in the thalamic units and the dSPNs, which disinhibit the thalamic units, leads to earlier onset times and, to some extent, lower thresholds (i.e., faster responses). The opposite effect was linked with the overall activity in GPi, FSIs, and cortical units, where greater activity leads to an overall slowing of responses by increasing the onset time and, to some extent, the boundary height. These opposing relationships are consistent with the inhibitory nature of projections from FSIs to dSPNs and from GPi to thalamus. In contrast, the differential activity between the two channels contributed relatively little to the responsiveness factor. It is worth noting that we display the inhibitory cortical neurons and FSIs in Fig 8 on both the sum and difference diagrams, but there is just one shared population in each of these regions, so the color-coding is the same in both cases.

The *pliancy* factor, on the other hand, was heavily dependent on the overall activity of indirect pathway regions across the two channels (Fig 8B). More activity in iSPNs, STN neurons (to some extent), thalamic neurons, and cortical inhibitory interneurons, as well as less activity in pallidal units, all associated with an increase of the boundary height and, to a slightly lesser extent, a decrease of the onset time. The former of these changes effectively increases the amount of information necessary to trigger a decision, while the change in onset time moderates the impact on reaction time somewhat. We also found a weaker positive association with overall firing rates in the direct pathway, specifically the dSPNs, along with the cortical units. As with the responsiveness factor, differential activity between channels had a more modest association with changes to the pliancy factor in the DDM process. The minor effects present again loaded more heavily onto indirect pathway nodes.

For the third factor reflecting action *choice* (Fig 8C), the situation reverses from what we see in the responsiveness and pliancy factors. Overall activity across channels had very little impact on the parameters of the DDM process related to the direction of information accumulation towards one action or the other. Between-channel differences in firing rates, however, had a robust association. Greater activity of dSPNs, GPe units, and thalamic neurons in channel A, compared to channel B, led to more positive drift rates and slightly lower starting bias factors. These effects result in a steeper angle of evidence accumulation towards the boundary representing action A, somewhat compensated by a starting point closer to action B. The opposite held for the indirect pathway units, with greater activity of iSPN, STN, and GPi units in channel A leading to lower drift rates and higher onset biases (i.e., angling the drift direction towards the boundary for action B while starting closer to the boundary for A). Put succinctly, the direction of the evidence accumulation process depends on the relative difference in activity across the action channels in all basal ganglia populations.

## Discussion and conclusions

The CBGT circuit provides a neural substrate that can control or tune how information is processed when a choice is being made among a set of available actions [5, 47]. After calibrating a computational model of the CBGT network to produce similar firing rates to those observed experimentally, we fit network decision-making behavior with a DDM model and performed CCA to reveal relationships among network synaptic weights and firing rates and the DDM parameters. This analysis demonstrates that different CBGT populations act as control ensembles capable of tuning aspects of behavior occurring during the decision-making process. The CCA describes these behavioral influences in terms of specific information processing factors that contribute to the action selection process, with the following manifestations in terms of the parameters of the DDM: (1) a responsiveness factor that highlights complementary changes in the onset time of evidence accumulation and, to a lesser extent, the boundary separation between actions, controlled largely by overall corticothalamic and direct pathway

activity; (2) a pliancy factor that represents antagonistic changes in boundary separation and in onset time, impacted primarily by overall indirect pathway activity; and (3) a choice factor that captures antagonistic changes in the drift rate towards one decision boundary and a shift in the starting position in the DDM domaixtn towards the other decision boundary, linked mostly with inter-channel differences in activity between populations. The algorithmic effects represented in each of these factors translate into impacts on behavior, including the vigor of action selection, observed in the decision-making process. Hence, our work provides a clear set of novel predictions about the behavioral impacts of activity changes within CBGT neuronal subpopulations.

These factors fit into a broader upwards-mapping conceptual framework [48] that underlies the approach taken in this study (see also [28, 49]). Specifically, by sampling the collection of synaptic weights ($W$) within the CBGT network (i.e., the *tuning* of the network), we varied the basis set by which *control ensembles* can drive characteristics of the *decision policy* observed in the network's overt behavior. In our framework, the parameters of the DDM ($P$) provide a description of the information processing dynamics of a given policy, but it was not clear *a priori* that the CBGT network can adjust its outputs ($R$) in a way that corresponds to variation of each individual DDM parameter. In practice, we observe that small sets of DDM parameters appear together within the components of the CCA that links firing rates to DDM parameters. We refer to these as the *factors* and they effectively parameterize the space of decision policies. The specific manifestations of factors from the dynamics of underlying control ensembles lead to decision policies with specific response time distributions and splits between A and B choices. These factors were obscured in a CCA applied directly to $W \rightarrow P$, likely because the intertwined pathway interactions in the CBGT circuit complicate the effect of weight changes on most firing rates in the network (e.g., S3 Fig) and the CCA ends up dominated by the clear-cut, direct effects of a few monosynaptic pathways. By performing CCA separately on $W \rightarrow R$ and $R \rightarrow P$—that is, the exploratory and hypothesis-focused evaluations, respectively—we were able to observe that the $R \rightarrow P$ mapping yields useful conclusions on the relationship between CBGT network properties and factors of decision policies. Interestingly, putting our $W \rightarrow R$ and $R \rightarrow P$ results together does not give a clear picture of which changes in synaptic weights would most effectively alter decision policies. Based on our analysis, we do not expect there to be simple CBGT synaptic weight modulation mechanisms that act directly to yield decision policy updates; rather, more subtle combinations of mechanisms, possibly including forms of neuromodulation, may perform this updating function.

The observation that specific control ensembles associate with specific factor configurations of the DDM provides a theoretical link between two disparate arms of the CBGT literature. On the one hand, there is the classical model of CBGT circuits as playing a critical role in action selection [47], through inhibition of competing action plans [5]. On the other hand, there is the emerging literature suggesting a critical role for CBGT systems in movement vigor [22, 23]. Our observations here posit a potential unification of these two models. Vigor in this context reflects the speed at which information (i.e., control signals) propagates once it enters the system. In the DDM, the drift rate, $v$, reflects the intensity with which information pushes the system towards one decision or another. All other DDM parameters reflect characteristics of the onset and offset of a decision process. Our analysis, as well as prior theoretical work [28], shows that the control ensemble that manages the drift rate works via direct competition between direct and indirect pathways. This finding is consistent with the observation in the vigor literature that selective stimulation of dSPNs or iSPNs increases or decreases, respectively, the vigor of an action [24]. Thus, the decision-making and vigor interpretations of CBGT circuit function both point to the competition between direct and indirect pathways as a critical control mechanism. We [19], and others [21, 50], have previously argued that this

competition reflects the intensity with which information accumulates through CBGT circuits, which would manifest as the speed of response times in decision tasks and movement vigor in motor control tasks, thereby unifying the two perspectives.

Our work also suggests that changes in firing rates in any of the neuronal control ensembles will change decision-related behavior in a way that corresponds to changes in the DDM parameters. Although changing all of the firing rates for neurons in each control ensemble together would provide the most complete impact on behavior, changes in individual population firing rates would be expected to push behavior in a similar direction, because no one measurement in our model CBGT network appears strongly in more than one factor. For example, based on the loadings for the responsiveness factor, decreases in overall GPi activity are predicted to have an effect on behavior that is consistent with adopting an earlier onset time for the evaluation of evidence in the DDM. A similar behavioral effect is expected from increases in overall thalamic activity and from each of the other rate changes indicated in the first column of Fig 8A. Results obtained from the pliancy and the choice factors, relating to overall indirect pathway activity and inter-channel differences in dSPN activity, respectively, agree with previous computational observations made with dopamine-related corticostriatal synaptic plasticity (see [51–53]), which showed that the boundary separation varies with overall iSPN activity while the drift rate varies with the inter-channel difference in dSPN activity [28] (see also reviews [19, 21, 50]). Interestingly, however, our new findings extend this analysis to encompass all of the classically recognized basal ganglia populations. In particular our findings indicate that differences across channels in all non-shared CBGT populations correlate with the DDM drift rate (and onset bias), while responsiveness and pliancy factors represent directions in DDM parameter space that are orthogonal to the action choice factor and are driven primarily by changes in onset times and boundary height. These choice ensemble dynamics are particularly relevant to the briskness with which decisions are made. The between-channel differences are not independent across all nuclei, of course, as differences in firing across A and B subpopulations in one nucleus will impact the relative firing of A and B subpopulations in all post-synaptic nuclei. This interdependency is reflected in the multiple nuclei with significant loadings on each of the control ensembles identified by the CCA. Presumably, dopaminergic effects, likely combined with other forms of neuromodulation, play a central role in tuning the activity levels of the control ensembles that we have identified. A natural direction for the extension of this work will be to implement this tuning process in various action-reward scenarios.

Our choice to use a spiking neuronal model tuned to match experimentally observed CBGT spike rates adds realism relative to more abstracted rate-based models, in which each network region is represented by a single, population-level firing rate. Yet it is possible, as in all modeling studies, that these different neuronal representations could lead to different conclusions. Despite methodological differences, our results on the influence of overall iSPN activity on DDM boundary height, as well as on the link between across-channel differences in dSPN activity and DDM drift rate, agree with our previous study focused specifically on striatal effects [45]. These results also agree with previous experimental work showing that inhibition of the striatum impairs information accumulation rate, analogous to the DDM drift rate, in mice [54]. These results represent a refinement of earlier findings based on DDM fits showing that modulations in general caudate nucleus activity associated with stimulus intensity and choice outcome in non-human primates can be interpreted in terms of evidence accumulation and choice bias [55]. Our prediction that the overall STN activity level, across action channels, modulates decision boundary height agrees with past work that linked a different spiking CBGT network model to the DDM, and also found that the level of STN activity correlated with boundary height [29]. Our study here yields the additional prediction that the difference

in STN activity levels between channels even more strongly relates to drift rate. This effect may not have been present in the previous model or may have emerged specifically in our work here due to our decision to use sums and differences in activity across channels as our firing rate dimensions, which may have revealed previously obscured distinctions. Future experiments that evaluate the impact of STN on decision-related behavior will be critical for testing the validity of these models' predictions. Interestingly, in this vein, recordings of local field potentials (LFPs) in human Parkinson's disease patients showed that STN LFP power, although a difficult quantity to relate directly to STN spiking, did correlate with decision thresholds in a direction consistent with both of these modeling studies [56]. Finally, we note that neither of these models takes into account the impact of voltage-gated ion currents and their properties in specific CBGT populations. Including these could add biological realism and, in theory, could alter our conclusions but also would introduce extra parameters that could interfere with mapping between activity levels and decision policy, and hence was not considered a useful step for this study.

For computational efficiency, we implemented some additional abstractions beyond the omission of voltage-gated currents. The current work does not include dopamine-influenced corticostriatal plasticity, since our aim was to vary network weights systematically, and future work will explore the weight modulations that actually result from know plasticity effects. Similarly, we consider only scenarios with unbiased evidence for the two choices, in order to derive the baseline upward mapping, but asymmetric evidence will be an important aspect of future plasticity studies. Another simplification, relative to physical experiments in biological systems, is that our CBGT network simulates the GPi output as always suppressing or blocking thalamic activity. Yet this relationship between the GPi and thalamus is more complicated. Evidence suggests that in non-human primates, GPi neurons and their thalamic targets tend to both increase their activity leading up to arm movement in a reaching task [57]. However, this counterintuitive observation may not represent the relationship between GPi and thalamic activity during the learning or decision-making process, since the experiments in this previous study were made for highly trained movements without a choice component. Another biological complication is that direct pathway signals and GPe outputs may not always inhibit GPi/SNr [58, 59], implying that more complex operational principles may be at play than those represented in our model.

The pathways that we included in our model here also represent only a subset of the known connections within the broader CBGT circuit. Other populations and pathways, such as arkypallidal neurons [60] or the cortico-STN (i.e., hyperdirect) pathway [29, 61–63], were not included in our simulations. These may play additional roles as parts of the control ensembles that we identified here, as well as possible new control ensembles, for example one contributing to a global inhibition, or stop factor, capable of preventing an action from being performed. We also opted for a primate variant of the CBGT pathways, with GPi used as the output nucleus of the basal ganglia, as opposed to the SNr, which is the primary output nucleus in rodents. It has been shown that SNr targets 42 different brainstem and midbrain regions, which may implement the direct motor impacts of basal ganglia activity, whereas SNr projections to thalamus may represent an efferent copy of the descending signal [64]. Even though our model does not consider this complexity of the output signals from the basal ganglia, in the future it could be modified to include a general brainstem motor target of the output nucleus, where a decision threshold would be imposed, along with one or more separate thalamic targets. If these were tuned similarly and the cortical inputs to the thalamic regions were sufficiently weak, then the model behavior would not be expected to change, despite the physical separation of the decision threshold from the feedback provided from thalamus to striatum and cortex. The weak influence of the cortico-thalamic synaptic weight on thalamic firing rate

shown in Fig 3 supports the likelihood of this robustness, although this would need to be tested to make sure. Increasing the biological realism of the model, however, should be a progressive goal across studies, particularly since our understanding of the anatomy and physiology of these circuits is still exponentially increasing with the advent of new experimental tools that give greater clarity on the underlying circuitry of these pathways.

Despite these limitations, our results clearly show that within the canonical CBGT circuits, specific subpopulations of cells contribute to specific aspects of the information processing capabilities of the network during decision-making. These low-dimensional relationships lead to very specific predictions for future experimental work. For example, global versus differential (i.e., between action channels) stimulation of direct and indirect pathways should manifest as changes in response speed and choice behavior, respectively. Stimulation of the indirect pathway alone, but globally across action representations, should result in contrasting effects on behavior, largely focused on threshold of evidence (i.e., pliancy), since the responsiveness factor is dominated by the direct pathway. These are just a few of the many experimental predictions generated by our results. In this way, the work described here can be seen as a guide for developing directed hypotheses for future experimental work.

## Methods

Overall, the goal of this work is to elucidate the complicated nonlinear mapping, call it $\mathcal{F} : W \rightarrow P$, from synaptic weights within the CBGT network to parameters in the DDM that are computed by fitting the distribution of choices and reaction times (RT) produced by various tunings of the CBGT network. The way that weight changes translate into different RT distributions is by impacting firing rates throughout the network. Moreover, firing rates are currently much more experimentally accessible than synaptic weights. Hence, we decomposed $\mathcal{F}$ into a composition of two relations: the association between firing rates and changes in DDM parameters, $\mathcal{H} : R \rightarrow P$, and the association between synaptic weights and network firing rates, $\mathcal{G} : W \rightarrow R$; that is, $\mathcal{F} = \mathcal{H} \circ \mathcal{G}$ (see Fig 1 for an overview). In this section, we describe the CBGT network model, the DDM model, and the data analysis that we use to link the two.

### Neural activity: CBGT network

In this study, we simulate behavioral data using a cortico-basal ganglia-thalamic (CBGT) network model adapted from previous works [20, 28, 49]. The network has been designed such that, when a stimulus is presented to the cortex, a decision between two different choices, A or B, is eventually made based on the firing rates of thalamic neurons, which are in turn impacted by neuronal firing throughout the network. The network consists of 9 different populations: the cortical interneurons (denoted by CxI) and the excitatory cortical neurons (Cx); the striatum, which includes the D1 and D2-expressing spiny projection neurons (dSPNs and iSPNs, respectively) and the fast-spiking interneurons (FSIs); the internal and prototypical-external globus pallidus (GPi and GPe, respectively); the subthalamic nucleus (STN); and the thalamus (Th). The model includes two groups of neurons in each population, one for the A channel and one for B, except for the CxI and FSIs, which are shared between the two. The neurons for each specific channel only project to other neurons in that channel except in a few instances; we include cross-channel connections from GPe to GPe, STN to GPi, and Th to Cx to reflect the divergence of the synaptic projections in these pathways [28]. The numbers of neurons per population are given in Table 1. The relative population sizes were taken from past work [20, 28]. The model includes more GPe neurons than SPN neurons, despite the relatively large size of the striatum, to allow for the divergence of connections from striatum to GPe [17, 65], and

we use the same number of STN neurons as GPe neurons to reflect the low divergence of connections from GPe to STN [66]. These larger numbers of GPe and STN neurons in the model also allow us to implement the relatively low connection probabilities between these regions [20, 67–71] (see Table 3 below).

Each neuron evolves according to the integrate-and-fire-or-burst model [20, 72] given by

$$C\frac{dV}{dt} = -g_L(V(t) - V_L) - g_T h(t) H(V(t) - V_h)(V(t) - V_T) - I_{syn}(t) - I_{ext}(t)$$

$$\frac{dh}{dt} = \begin{cases} -h(t)/\tau_h^- & \text{when } V \geq V_h \\ (1 - h(t))/\tau_h^+ & \text{when } V < V_h \end{cases}$$

where the equation for the membrane potential $V(t)$ includes a leak current with conductance $g_L$ and reversal potential $V_L$; a low-threshold $Ca^{2+}$ current with maximal conductance $g_T$, reversal potential $V_T$, gating variable $h(t)$, Heaviside step function H, and time constants $\tau_h^+$ and $\tau_h^-$ representing the rate of change of the gating variable $h(t)$ before and after the membrane potential reaches a certain constant voltage threshold $V_h$, respectively; a synaptic current, $I_{syn}(t)$; and an external current $I_{ext}(t)$.

The synaptic current itself includes excitatory *AMPA* and *NMDA* and inhibitory *GABA* components, such that

$$\begin{aligned} I_{syn} &= g_{AMPA}s_{AMPA}(t)(V(t) - V_E) + \frac{g_{NMDA}s_{NMDA}(t)(V(t) - V_E)}{1 + e^{-0.062V(t)/3.57}} \\ &+ g_{GABA}s_{GABA}(t)(V(t) - V_I), \end{aligned}$$

where each $g_i$ denotes the maximal net channel conductance for $i \in \{AMPA, NMDA, GABA\}$. $V_E$ and $V_I$ are the reversal potentials for excitation and inhibition, respectively, and $s_i(t)$ are open channel fractions, with dynamics given by

$$\frac{ds_{AMPA}}{dt} = \sum_j \delta(t - t_j) - \frac{s_{AMPA}}{\tau_{AMPA}},$$

$$\frac{ds_{NMDA}}{dt} = \alpha(1 - s_{NMDA})\sum_j \delta(t - t_j) - \frac{s_{NMDA}}{\tau_{NMDA}},$$

$$\frac{ds_{GABA}}{dt} = \sum_j \delta(t - t_j) - \frac{s_{GABA}}{\tau_{GABA}},$$

for spike onset times $t_j$, rate constant $\alpha$, and decay rates $\tau_i$ for each choice of $i$. Note that because NMDA has the slowest decay rate, we design the $s_{NMDA}$ equation to explicitly prevent this variable from exceeding one.

The external current, which is used to tune the baseline firing rate of each population, is given by

$$I_{ext}(t) = S_{ext,AMPA}(V(t) - V_E) + S_{ext,GABA}(V(t) - V_I)$$

where $S_{ext,X}$ is a mean-reverting random walk depending on the external input frequency, the efficacy of the external connections, and the number of external connections (see S1 Table).

The neurons within a population that correspond to the same choice are connected to each other and, in some cases, to those in FSI and CxI, resulting in two different channels, which we call channel A and channel B. To simplify notation, we will label a specific population with the subscript A or B if we want to specify that we consider the A-channel sub-population or the B-channel one, respectively. The established synaptic pathways between regions that we include

in the model can be found in Fig 1 (upper panel). In all simulations, the synaptic conductances for the two channels are identical. Within each channel, the connections from *Cx* to *dSPN* and from *Cx* to *iSPN* are equal in our control parameter set, but we allow these to differ in subsequent simulations.

While Fig 1 (upper panel) shows the pathways between populations, the individual neurons within populations are connected with specified probabilities and synaptic conductances *g* in *nS* (which we also call weights). The connection probabilities and weight ranges considered, which we calibrated to obtain similar firing rates to those observed experimentally [32–40] during resting states (baseline) and decision processes (Table 2), are presented in Table 3.

To represent the presentation of a stimulus to the network, we increase the external input frequency for $Cx_A$ and $Cx_B$ from 2.2 *Hz* to 2.5 *Hz*. Subsequently, both channels compete to make a decision. If the thalamic firing rate associated with channel A reaches a specific threshold, which we take as 30 *Hz*, before the thalamic firing rate associated with channel B, then we say that decision A is selected, with a similar condition for decision B. We define the time from the stimulus onset to the decision as the *reaction time* for a trial. If the decision thresholds for both channels are not reached in a time window of 800 *ms*, then we say that no decision has been performed and we end the trial.

Unlike [49, 53], in the version of the network implemented in this work, spike-timing-dependent plasticity from the cortex to the spiny projection neurons is not included. The rest of the structure and parameters remain as in [49] except for two changes. First, the leak conductance of thalamic neurons, $g_L$, is decreased from 25 to 18 *nS* to fit experimental information about their firing rate under baseline conditions. Second, after a decision is made, the external input presented to cortex as a stimulus is maintained for 300 *ms*, as in the earlier work, but here we only maintain this input at 75% of its original strength.

## Network specification and network behavior

By modifying the values of the maximal conductances or weights *g* of the connections from one population to another, we obtain different configurations of the network that we call the network *tunings*. Given the large number of weights in the full model, for tractability, we only explore variations across the main feed-forward connections in the CBGT network; in simulations, we also see that these connections have the strongest influence on reaction times. The weights for the rest of the connections in the network are kept constant. Specifically, we fixed intra-cortical connection weights that are not specifically part of the CBGT control system; recurrent weights within the striatum, which we generally found to have a relatively weak impact on behavioral data; GPe-GPe weights, to which behavioral data was highly sensitive; and several NMDA connection strengths. Therefore, each tuning is generated by considering different values of the 14 specific connections highlighted in bold in Table 3 (1st column).

For each varied connection type, we identify the interval of weights over which the population firing rate remains in our allowed range, thus establishing upper and lower bounds on the *g* values to be considered for that connection (see Table 3, 4th and 5th columns).

We performed Latin hypercube sampling (LHS) on values of the 14 varied weights to randomly specify *N* = 300 different weight configurations, and thus 300 unique network tunings. In brief, for each connection, we partitioned the allowed range into 300 bins of the same size. For each of our 300 iterations, we selected one bin uniformly at random for each weight, independently across weights. These selections were made without replacement, so that each bin was used exactly once, resulting in 300 multi-dimensional bins. Within each multi-dimensional selected bin, we randomly specify 14 weight values that together correspond to one

configuration of the network. To perform the LHS we use the lhsmdu function implemented in Python.

Finally, for each tuning, we simulated 300 decision trials, each of which ended when one of the thalamic subpopulations reached the decision threshold or when the allowed time for the decision expired, if no decision had been reached by that time; note that the use of 300 for both the number of trials and the number of LHS bins is inconsequential. On each trial on which a decision was made, we considered as the CBGT output the reaction time from the introduction of the stimulus to the decision, the average firing rates of all populations over that time period, and which option was selected.

## Drift-diffusion model

For each tuned network, we fit the simulated behavioral data from the CBGT network, specifically the reaction time and the choice made on each trial, with the drift-diffusion model (DDM), a stochastic model that instantiates the statistically optimal performance on two-alternative decision tasks (see [28, 73, 74]). The DDM assumes that decisions are made by an accumulative stochastic process. Given two different boundaries, one for each choice, which are separated by a distance called the decision threshold ($a$), and given also a starting bias $za$ determined by multiplying the decision threshold by a bias factor ($z$), the accumulated evidence $\theta$ is a function of time $\tau$, defined from $\tau = 0$ up to a stopping time that occurs when $\theta$ reaches one of the two thresholds, $\theta = 0$ or $\theta = a$. The behavior of $\theta$ is governed by the following stochastic differential equation and starting condition:

$$d\theta = vd\tau + \sigma dW \quad \text{if } \tau > t$$
$$\theta = za \qquad\qquad \text{if } \tau \leq t$$

where $v$ denotes the rate of evidence accumulation, $t$ is the time of onset of evidence accumulation, and $\sigma$ represents the level of noise in the process, which is given by the standard deviation of a white noise process $W$, taken in our simulations to be 1. For each trial, the choice and the reaction time for the DDM are determined by which boundary is reached and the time elapsed between the onset time $t$ and the moment when $\theta$ reaches one of the boundaries, respectively.

In our simulations, for each tuned network, we estimate the values of the quadruple ($a$, $v$, $t$, $z$) for which the DDM behavior best fits the reaction times and the selected choices obtained with the CBGT network. For this purpose, we use the Hierarchical Drift-Diffusion Model (HDDM) implemented in Python (see [31]) using as inputs both the reaction time (RT) and the specific decision made on each successful trial (i.e., distinguishing between A and B), after first removing the trials where no decision was made.

## Data analysis

We consider three different sets of data: the weights selected that define the various CBGT network configurations, $W$; the averaged firing rates obtained from all populations of the CBGT network, $R$, computed from the time of stimulus presentation to the moment that thalamic activity reaches the decision threshold; and the DDM parameter values obtained using the HDDM, $P$. For clearer visualization and analysis of the results, we considered the averaged firing rates of each CBGT population summed across both channels, A and B, and the difference in the averaged firing rates of the populations between the two channels. Since there are 7 populations in each channel plus two populations that the channels share, $R$ is a $300 \times 16$ data table, while the dimensions of $W$ and $P$ are $300 \times 14$ and $300 \times 4$, respectively. Once we form these matrices, we perform additional analyses to investigate relations between them.

**Canonical correlation analysis.** We use canonical correlation analysis (CCA, [46]) to infer the relation between the pair of data sets $(W, R)$ and again to infer information relating the pair of data sets $(R, P)$. In general, given two data sets $\mathbf{X} = (x_1, \ldots, x_n)^\top$ and $\mathbf{Y} = (y_1, \ldots, y_m)^\top$, CCA searches for vectors $\mathbf{u} \in \mathbb{R}^n$ and $\mathbf{v} \in \mathbb{R}^m$ such that the correlation between $\mathbf{u}^\top \mathbf{X}$ and $\mathbf{v}^\top \mathbf{Y}$ is maximal. The vectors $\mathbf{u}$, $\mathbf{v}$ are the first pair of canonical variables. Subsequently, for each $i = 2, \ldots, N$, we compute the $i$th pair of canonical variables by maximizing the same correlation subject to the restriction that the new pair is uncorrelated with the previous $i - 1$. We call $N$ the *number of components* and select it to be less than or equal to $\min\{n, m\}$. To compute the CCA, we use the *CCA* function of the *sklearn* package in Python. To discern the minimum number of components necessary to fit the data, we consider the number of components providing the best $R^2$ score for the predicted data, which is computed for the *CCA* function [75]. This score is defined as $1 - a/b$ for

$$a = \sum_{i=1}^{m} \left(y_i - y_{pred_i}\right)^2, \quad b = \sum_{i=1}^{m} \left(y_i - \bar{y}\right)^2$$

where $y_i$ are the entries of the data matrix $\mathbf{Y}$, $\bar{y}$ is the mean of the $y_i$, and $y_{pred_i}$ is the predicted value for the $i$th data element from the dimension reduction learned during the CCA. This expression quantifies the extent to which the CCA represents the data, relative to chance. The best possible fit would yield an $R^2$ score equal to 1; notice that, although a 0 score represents a poor fit, the score can in fact be negative, meaning that the model based on CCA is worse than would be expected by chance.

To ensure that we do not obtain spurious results based on our data sample, we apply a 4–fold cross-validation analysis. In this step, we split the 300 tuned networks into two different blocks: the training set, containing 75% of the tuned networks, and the testing set, comprising the remaining 25% of the data. To achieve this splitting, we order the tuned networks from 1 to 300 and then in each $i$-fold, with $i \in \{1, 2, 3, 4\}$, we consider the tunings in positions $(75(i - 1), 75i]$ as the testing set and the rest as the training set. Hence, for different numbers of total components, we compute the CCA using the training set and we test the resulting model using the testing set, from which we compute the CCA score.

Conceptually, the canonical variables expose the variable groupings that most significantly contribute to correlations between specific data sets. However, they are subject to variability across samples and can be highly affected by multicollinearity. Hence, to interpret the CCA results, we consider canonical loadings, which represent the correlations between the original variables and the canonical variables.

**K-means clustering.** To identify different clusters of data in the HDDM output, we performed $K$-means clustering using the *KMeans* function of the *sklearn* package in Python. For each tuned network, we considered the point in $\mathbb{R}^4$ given by the $(a, v, t, z)$ obtained from DDM; together, these comprise a data set of 300 points. We separated these data into $K$ different clusters such that each observation lies in the cluster with the nearest mean, minimizing the variance within clusters.

Before using the clustering technique, we performed a sensitivity-based normalization of the HDDM output $(a, v, z, t)$. In this procedure, each component $x \in \{a, v, z, t\}$ is modified as

$$x \leftarrow (x - \bar{x})\Delta_x$$

where $\bar{x}$ refers to the mean of $x$ over the different tuned networks and $\Delta_x$ is the centered

difference formula to compute the change in the reaction time relative to the change in $x$, that is

$$\Delta_x = \frac{RT(x+h) - RT(x-h)}{2h},$$

where $h = 0.1$ and $RT(y)$ is the mean reaction time obtained after running the HDDM $10^5$ times with input variable $y$ and fixed values of the components in $\{a, v, z, t\} \backslash y$.

To determine the optimal number of clusters into which to split the data, we subject the data to $K$-means clustering for each $K \in \{1, 2, \ldots, 10\}$. In each case, we compute the mean of the silhouette coefficients over samples. The silhouette coefficient is a measure of the proximity of each cluster point to neighboring cluster points. This coefficient is calculated as $(b - a)/\max\{a, b\}$ where $a$ denotes the mean distance between points within a cluster and $b$ is the distance between a data point and the nearest cluster to which it does not belong. This calculation yields a value lying in $[-1, 1]$, where a value close to 1 indicates that the specific sample is not close to the neighboring clusters and hence it has been assigned to the correct cluster, a value close to 0 indicates that the specific sample is very close to or on the boundary defining the different clusters, and negative values close to $-1$ indicate that the specific sample could be wrongly assigned. Hence, higher values of the mean silhouette coefficient indicate a better clustering performance. We compute the silhouette coefficient using the *Silhouette_-score* function in the *sklearn* package in Python.

The rest of the parameters used in the $K$-means clustering process are standard values of the *KMeans* function in the *sklearn* package in Python. That is, 10 repetitions of the $K$-means algorithm are run using different centroid seeds, with a maximum of 300 iterations per run. To check for convergence of the method, the Frobenius norm of the difference in the cluster centers of two consecutive iterations is evaluated by the KMeans function itself against a relative tolerance of $10^{-4}$. The $K$-means problem is solved using Elkan's algorithm with precomputed distances, 0 verbosity mode and "None" as the random state.

## Supporting information

**S1 Fig. Firing rate histograms across tuned networks.** Each panel depicts, for each neuronal population, the histogram of the mean firing rates from stimulus to decision across the 300 different tuned networks.
(TIFF)

**S2 Fig. 3-means clustering results.** The first row shows the two main clusters of DDM parameter sets obtained after applying K-means clustering with $K = 3$. The third cluster consisting of just a few points with high $a$ values is not shown here. The next four rows present the histograms of the DDM parameters $a$, $v$, $z$ and $t$, respectively, corresponding to the cluster at the top of each column. The bottom row represents the reaction time histograms for each cluster.
(TIFF)

**S3 Fig. Canonical loadings obtained after applying the CCA to the pair *W*, *R*.** Relation between the different synaptic weights and the firing rate of each population in the CBGT network. Panels A, B and C stand for the first, second and third components of the CCA. 1st column subpanels correspond to the weight difference between channels. 2nd column subpanels correspond to the overall weights across the two channels. The loadings of the weights, shown via color-coding of synaptic pathways in the image (1st and 2nd columns) are quite weak overall; note the green pathway from GPi to thalamus in the first component (2nd column), the

green pathway from GPe to STN in the second component (2nd column), and the green (magenta) pathway from GPe (STN) to GPi (2nd column). The 3rd column subpanels show the channels' overall firing rates while those in the 4th column depict the difference in rates between the A and B channels. The color-coded loadings of the firing rates are also strong in a only a relatively small number of sites in each CCA component.
(TIFF)

**S4 Fig. Goodness of fit in the DDM model.** DIC values for the DDM parameters used to fit behavioral data from 300 networks tuned with different weights. The dashed horizontal line represents the DIC value equal to 0. The solid grey curve separates the two different clusters shown in S2 Fig. Data corresponding to the cluster with high $t$ values (right panels in S2 Fig) is represented below the curve using circle markers while that corresponding to low $t$ values (left panels in S2 Fig) is represented above the curve using square markers. Dots in the third cluster (not shown in S2 Fig) with reaction times in [20, 200] $ms$ are plotted in grey using triangle markers. The color coding in each panel corresponds to the values of a specific DDM parameter, indicated above the color bar, in the fits.
(TIFF)

**S1 Table. External current parameters used to simulate the external input arriving at the various CBGT populations.** From the first column to the last, we specify the receiving population, the receptor type of the external current, the frequency of the external input, the efficacy of the specific external connection, and, finally, the number of external connections projecting to the population. The time decay constant is $\tau = 2$ $ms$ for the AMPA receptor and $\tau = 5$ $ms$ for the GABA receptor.
(PDF)

## Acknowledgments

We thank Fred Hamker, David Robbe, and Eric Yttri for helpful comments on this work.

## Author Contributions

**Conceptualization:** Jonathan E. Rubin, Timothy Verstynen.

**Data curation:** Catalina Vich, Matthew Clapp.

**Formal analysis:** Catalina Vich.

**Funding acquisition:** Catalina Vich, Jonathan E. Rubin, Timothy Verstynen.

**Investigation:** Catalina Vich, Matthew Clapp, Jonathan E. Rubin, Timothy Verstynen.

**Methodology:** Catalina Vich, Jonathan E. Rubin, Timothy Verstynen.

**Software:** Matthew Clapp.

**Supervision:** Jonathan E. Rubin, Timothy Verstynen.

**Validation:** Jonathan E. Rubin, Timothy Verstynen.

**Visualization:** Catalina Vich, Jonathan E. Rubin, Timothy Verstynen.

**Writing – original draft:** Catalina Vich, Jonathan E. Rubin, Timothy Verstynen.

**Writing – review & editing:** Jonathan E. Rubin, Timothy Verstynen.

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
