## [Decision Letter · Decision Letter 0]

7 Mar 2022

Dear Dr. Verstynen,

Thank you very much for submitting your manuscript "Identifying control ensembles for information processing within the cortico-basal ganglia-thalamic circuit" for consideration at PLOS Computational Biology. As with all papers reviewed by the journal, your manuscript was reviewed by members of the editorial board and by several independent reviewers. The reviewers appreciated the attention to an important topic. Based on the reviews, we are likely to accept this manuscript for publication, providing that you modify the manuscript according to the review recommendations.

Your manuscript was reviewed by three reviewers. All of them have appreciated the work but have also found places where the manuscript can be improved. One of the concern is the biological relevance of the work i.e. to what extent the results of the model are applicable in a biologically plausible setting. This requires more justification for the model. Another concern is that whether bimodaltiy is a property of the model or of the fitting procedure. These concerns needs to be addressed rigorously. More details comments from the reviewers are appended below.

Sincerely,

Arvind Kumar, Ph.D.

Guest Editor

PLOS Computational Biology

Kim Blackwell

Deputy Editor

PLOS Computational Biology

[LINK]

Your manuscript was reviewed by three reviewers. All of them have appreciated the work but have also found places where the manuscript can be improved. One of the concern is the biological relevance of the work i.e. to what extent the results of the model are applicable in a biologically plausible setting. This requires more justification for the model. Another concern is that whether bimodaltiy is a property of the model or of the fitting procedure. These concerns needs to be addressed rigorously. More details comments from the reviewers are appended below.

Reviewer's Responses to Questions

**Comments to the Authors:**

Reviewer #1: This paper seeks to link two levels of modelling of decision making, namely detailed biophysical models of the basal ganglia circuit and abstract diffusion model. The relationship between these models was numerically studied in the past by the Authors as well as by Ratcliff & Frank (2012), but this paper goes beyond these earlier studies and present a very comprehensive analysis. The Authors simulate the choices with the basal ganglia models with many different sets of parameters, and then estimate parameters of the diffusion model from these simulated choices. Subsequently they use nice statistical methods to describe how parameters of biophysical models are related to diffusion model parameters.

I have to admit that I have mixed feelings about this paper. On one hand, it is exceptionally clearly written, and nicely illustrated with very informative figures. All numerical analyses employ sophisticated and appropriate methods.

On the other hand, I ask myself in what ways this analysis can be useful. What can the readers gain from such analysis? Although the paper provides some motivation and experimental predictions, I think more justification for this work would be useful in the paper.

Additionally, one can question to what extent the conclusion on the relationship of the basal ganglia model to diffusion model would generalize to relationship between biological basal ganglia to diffusion model. Although Authors used a well-developed model, presumably using a different model could give different results (e.g. Ratcliff & Frank got different conclusions with their version of the basal ganglia model). Discussion of this issue would be useful.

Despite these limitations, the manuscript provides a nice methodology for numerically linking models of different levels of abstraction, that could possibly be applied in other domains.

Minor comments:

Line 435 – It would be useful discuss the experiments on linking the neural activity in the basal ganglia with parameters of diffusion model, and to compare if the conclusions from the models match with them (e.g. Herz et al. 2017 eLife).

Line 582: Over what time interval, the firing rates are averaged to compute R? Is it until reaching the decision threshold, or over the whole 800ms episode?

Reviewer #2: In their manuscript “Identifying control ensembles for information processing within the cortico-basal ganglia-thalamic circuit”, Vich et al. examine how networks including the cortex, thalamus and the basal ganglia contribute to decision making. Using canonical correlation analysis they examine how changes in the synaptic weights of a network model relate to the parameters of drift diffusion models. This is important because it connects two different levels of descriptions, physiologically-grounded networks and abstract models of choice behaviour. The authors find that certain connections and pathways in the basal ganglia affect specific DDM parameters and can be interpreted in terms of responsiveness, pliancy and choice.

I think this is a very strong paper with interesting results and convincing approach. To me the key insights include that specific connections and pathways in the basal ganglia can be related to DDM parameters, which can be estimated from behavioural data. While such relations have been speculated to exist before, the authors provide a model description to demonstrate this. I think this makes a lot testable predictions for studies combining electrophysiology with choice tasks. The paper is written very well and easy to read. I have several questions, comments and thoughts that the authors can consider to further improve the manuscript.

1. (Figure 4) It wasn’t clear to me why the data was sorted according to the order in which the trials were simulated (as these are independent anyway) and the firing rates shown don’t seem to convey much information. Instead the firing rates could be grouped by the decision taken (eg a block on the left for decision=0 and one on the right for decision=1). Within each block the trials could then be sorted by RT so that a clearer relation between the firing rate and the behavioural output might be visible.

2. (lines 184-185 & Figure 4) Did the authors consider the goodness of fit for the comparison of decision outcomes and reaction times between the CBGT and DDM models? There might also be some regions of the parameter space where the fits just don’t work very well.

3. (lines 273-276) What was the reasoning for grouping the firing rates in this way? Obviously, taking the difference in firing rate between channels lead to interesting results as reported, but my intuition would have been to start with the firing rates averaged within each channel, separately for both decisions. Did the authors consider alternative firing rate groupings and could they add a rationale for their approach here? Related to this, for Figure 2 I was wondering why the choice-specific averages are not shown? This would also avoid the mentioned issue that the GPi decreases are barely visible.

5. Throughout the manuscript the authors examine the scenario with the evidence for A and B being the same. What is the reason for looking at that scenario exclusively, and not consider cases where the decision is biased?

6. In the model analysis, changes in the weights are considered. However, the number of neurons in each population and their connection probabilities also have a large effect on the network activity. What were those parameters based on, e.g. taken from a previous model and/or based on experimental studies (e.g. in Table 1 and 3)? This might e.g. affect the GPe/STN interaction and whether weight changes manifest as a net excitation or inhibition in GPi.

7. (line 241) It is mentioned that the 2nd CCA was to address the ‘primary hypothesis of interest’. I don’t think it is stated anywhere what that hypothesis is.

8. In Figure 5 the authors show that in their random sampling of network weight parameters two main clusters emerge. As far as I understand it, the CCA analyses are also based on the random sampling of 300 network configurations. I was wondering to what degree the overall results, ie the mapping between R and P depends on the sampling. For example, network configurations belonging to the ‘High a and low t’ cluster might be systematically different in their activity patterns than network configurations belonging to the other cluster. In the CCA analysis network configurations from both clustered are included by the random sampling. Could this affect the results in terms of the found loadings? Would the loadings be similar if the CCA analysis was performed for network configurations from each cluster separately?

9. (line 367): the authors mention that the CCA applied directly to W->P did not work as well. While reading your manuscript I was also wondering at an earlier section about why you did not just use W->P directly. What are possible reasons for why that approach does not work well? Above (l. 447) you had motivated your approach by firing rates being experimentally more accessible, but then here (l 367) it just sounds like that approach did not work well, so why is that?

Minor points / suggestions:

Figure 5: I would normalise the histograms (eg to %) instead of raw frequency so that it is easier to interpret independent of the number of trials run in the simulations. In panels C and E I was wondering whether the data partly clustered at the lower end of the allowed parameter range? Or were a and t allowed to fall below 0.25 and ~0.29 and the high frequency there is not due to a lower cap?

Figure S3: I struggled to understand the 3rd and 4th column - what is the interpretation for the color in a given region?

Figure 6: As the components can only be integer numbers, I would not show the x-labels & tick marks for fractions (e.g. “2.5”) for clarity. I was wondering initially why only up to 4 components are shown in panel B, but I think the reason is that the 2nd CCA is limited by the number of components chosen for the 1st CCA? If so, this might be helpful to mention somewhere.

line 289: This is rather subjective, but the term ‘pliancy’ didn’t seem very intuitive to me for describing the 2nd factor. Doesn’t it somewhat correspond to what has also been referred to as ‘urgency’ in the work of Paul Cisek and others, e.g. here https://doi.org/10.1523/JNEUROSCI.1844-09.2009 ?

For the understanding of Figure 7 and 8 it might be good to clarify somewhere that they actually contain the same data and are two different visualizations of the loadings (if I understood it correctly…)

lines 150-252: Not clear which studies and data sets you are referring to, please add some references.

line 305: unclear, in what sense is the cortical impact more ‘subtle’?

line 377: I would suggest to phrase this a bit more cautiously, as some people (like me) would also consider e.g. arkypallidal and prototypical neurons to comprise ‘major basal ganglia populations’.

line 386: Dopamine effects are mentioned here. I think even without considering other scenarios or explicitly modelling dopamine neuromodulation, you might already be able to make some predictions on the effect of dopamine, based on the simplified view that dopamine would overall excite D1 SPNs and inhibit D2 SPNs. In particular, the dSPN effect on responsiveness would indicate that D1 receptor activation reduces t and a; D2 receptor activation reduce a but increase t; and dopamine release in general would increase v and decrease z. As there are already many studies on the effects of dopamine on choice behaviour you might draw some interesting connections there.

lines 27-29: “D1-expressing spiny projection neurons (SPNs) in the striatum, unleashes GABAergic signals that can suppress activity in the CBGT motor output nucleus (internal segment of the globus pallidus, GPi, in primates or substantia nigra pars reticulata, SNr, in rodents)”.

I would rephrase this statement, as it seems to suggest that rodent SNr is the equivalent of primate GPi, or that primates don’t have an SNr.

Best wishes,

Robert Schmidt

Reviewer #3: This manuscript's goal is to understand more deeply the roles of of cortico-BG-thalamic (CBGT) components in decision-making tasks, by continuing a great line of work mapping the complex CBGT circuit to tractable, simpler models. The approach taken here is to set up a complex spiking-neuron CBGT model whose firing rates fall within reasonable ranges; fit a DDM model to reaction times and choices made by the model in response to identical inputs for two choices; repeat for 300 sampled weight sets; then use CCA to study how changes in firing rates correlate with changes in DDM parameters. The CCA analysis revealed a mapping between three key elements of decision making the activity shared among specific components of the CBGT circuit. The manuscript clearly reports a substantial body of work, with a clear goal, some solid, interesting results, and a novel use of CCA (to me, at least).

Main suggestion below are towards further refining the paper to make elements clearer for the reader, and head off some potentially nagging concerns they may have:

- Introduction lines 46-62): whole paragraph on action selection vs vigor, followed by claim that CBGT's proposed roles in action selection and vigour can be reconciled by understanding how the pathways within the CBGT process information. This is not referenced again in the paper, and is unclear how the present work address the issue. Seems tangential to the main aim of getting a handle on how CBGT circuit maps to decision making

- Results:

- A major result is that the DDM fits fall into two clusters, suggesting two main parameterisations of decision-making in this CBGT model, based only on variation in boundary height and onset (lines 218-222). How do we know this bimodality is a real property of the CBGT model and not a consequence of the DDM fitting procedure? One could imagine that the DDM fitting procedure has two local minima, corresponding to each of these modes, such that re-running the fitting would put a given CBGT model in a different cluster.

- Given that there are two clusters of fits, how do we then interpret the CCA analysis of (R->P), when CCA measures linear correlation but the DDM parameters are discontinuous?

- how do we read the relationship between the canonical loading matrices in Figure 7A and Figure 7B? This relationship underpins all the main conclusions on the mapping from CBGT models (Fig 7A) to the decision making process (Fig 7B) in lines 282 onwards, but the connection between the two matrices is never made explicit. For example, what are the canonical loadings measuring (i.e. why do we care about the correlation between the canonical vectors and the original data?) And given that we are comparing between pairs of like-numbered columns in Figs 7A and 7B, perhaps explain how we should make that comparison - presumably a canonical loading on one axis (e.g u) imply a strong relationship on the other (e.g. v) for a given mode?

- Discussion could be improved in places:

- a W->P mapping is mentioned, but is not in the Results

- perhaps join the dots for the reader: what do we learn from W->R and R->P mappings together? I.e. what changes in synaptic weights would alter DDM models? Or do we need the W->R mapping at all?

- predictions: there seem to be some quite deep predictions of how causal interventions in the CBGT loop would change decision making parameters in experiments - I'd encourage a clear explicit account here, to aid the interested experimentalist

Minor suggestions/queries for text and figures:

- Abstract: it is not clear what "responsiveness" and "pliancy" are until deep into the paper

- Figure 1: legend says three decision making trials, but there seem to be four pairs of vertical black/green lines? Would also be useful to plot the decision threshold (30 Hz) on the thalamus output panel

- line 144-145 "varying... specific connections" - would help reader here to have some idea of which connections are being sampled, to summarise Table 3

- line 153 "each permuted network" - unclear what was permuted? "Sampled" meant here?

- Figure 3: perhaps draw horizontal line at 0, to make clear the sign of each coefficient (and mapping to synapse's sign)

- lines 169-170 and Figure 3: claim that many populations are insensitive to variation of synaptic weights. As a linear model was used here, is it more accurate to say the firing rates are not monotonic function of the variation in weights?

- lines 181-185, "The behaviour..." onwards: perhaps delete as this was rather confusing, and then explained in full anyway from lines 197 onwards.

- lines 188-194: perhaps explicitly give the RTs predicted by the model, which would seem to be about 150-350 ms given standard "execution time", then give citations to papers reporting these RTs

- Figure 4: implies all model outputs are used to fit DDM model. Perhaps plot intermediate step of choice and RT distributions, and show DDM fitted from these

- lines 207-210: suggestion here that drift rate and bias are normally distributed - true, or just unimodal?

- Figure 5: could not see the z parameter variation along the drift axis in this 3D plot. Perhaps give two 2D colour-coded plots to show main directions of change referred to in the text.

- lines 252 "in many empirical neurobiology datasets" - citations here?

- lines 284-280 "grouped in two ways" and "the second CCA model". Took me a while to figure out that the "second CCA model" was referring to R->P, and not to the second way of grouping the firing rates. Perhaps disambiguate.

- Figure 7: define the colour code.

- Methods, Table 1: why are GPe and STN an order of magnitude bigger than the other structures in the model, when they are orders of magnitude smaller?

- Methods, Table 2: if this is a primate model, explicitly linked to human performance in the Results, what's the rationale for data taken from rats here?

- Methods, Table 3: STN-GPi, Th-Cx, and Th-CxI only have NMDA receptors listed, when they all have AMPA too as far as I know. Part of the model, or just an omission of a detail?

- Methods, DDM model fitting: what noise level (sigma) was used for the fits?

- Methods, CCA: how was R^2 computed? (a) what are y_true and y_pred for CCA - I'm unclear on how the comparison is being made between "true" data and predicted: initially I thought these respectively the canonical variates (i.e. u'X and u'Y) computed on the test data? (b) how was it computed over multiple components? As an average over the components, or by concatenating the predicted and true vectors, or...?

Mark Humphries

**Have the authors made all data and (if applicable) computational code underlying the findings in their manuscript fully available?**

Reviewer #1: Yes

Reviewer #2: Yes

Reviewer #3: Yes

PLOS authors have the option to publish the peer review history of their article (what does this mean?). If published, this will include your full peer review and any attached files.

Reviewer #1: No

Reviewer #2: **Yes: **Robert Schmidt

Reviewer #3: **Yes: **Mark Humphries

Figure Files:

Data Requirements:

Reproducibility:

References:

---

## [Decision Letter · Decision Letter 1]

27 May 2022

Dear Dr. Verstynen,

We are pleased to inform you that your manuscript 'Identifying control ensembles for information processing within the cortico-basal ganglia-thalamic circuit' has been provisionally accepted for publication in PLOS Computational Biology.

Best regards,

Arvind Kumar, Ph.D.

Guest Editor

PLOS Computational Biology

Kim Blackwell

Deputy Editor

PLOS Computational Biology

Dear Dr. Verstynen

I am pleased to inform you that the three reviewers are happy with your revisions and the manuscript can be accepted for publication pending some formalities of formatting etc. Many congratulations.

Best, Arvind

Reviewer's Responses to Questions

**Comments to the Authors:**

Reviewer #1: The Authors satisfactory addressed my comments.

Reviewer #2: the authors have addressed all my comments well

Reviewer #3: The authors have responded thoughtfully to all comments, and made appropriate changes to the manuscript. The manuscript is suitable for publication.

It was reassuring that the repeated fitting to sample models from the two clusters of DDM fits found the same results. Their point about the AMPA receptors having little impact in high firing-rate regimes, and so why they report only NMDA receptors in the model, is an interesting little nugget.

**Have the authors made all data and (if applicable) computational code underlying the findings in their manuscript fully available?**

Reviewer #1: None

Reviewer #2: Yes

Reviewer #3: Yes

PLOS authors have the option to publish the peer review history of their article (what does this mean?). If published, this will include your full peer review and any attached files.

Reviewer #1: No

Reviewer #2: **Yes: **Robert Schmidt

Reviewer #3: **Yes: **Mark Humphries

---

## [Editor Report · Acceptance letter]

17 Jun 2022

PCOMPBIOL-D-22-00117R1 

Identifying control ensembles for information processing within the cortico-basal ganglia-thalamic circuit

Dear Dr Verstynen,

I am pleased to inform you that your manuscript has been formally accepted for publication in PLOS Computational Biology. Your manuscript is now with our production department and you will be notified of the publication date in due course.

With kind regards,

Agnes Pap
